# A Self-Representation Learning Method for Unsupervised Feature Selection using Feature Space Basis

**Prayag Tiwari**  *prayag.tiwari@ieee.org*
*School of Information Technology, Halmstad University, Sweden*

**Farid Saberi-Movahed**[*]  *f.saberimovahed@kgut.ac.ir*
*Department of Mathematics, Faculty of Sciences and Modern Technologies,*
*Graduate University of Advanced Technology, Kerman, Iran*

**Saeed Karami**  *s.karami@iasbs.ac.ir*
*Department of Mathematics, Institute for Advanced Studies in Basic Sciences (IASBS),*
*Zanjan, 45137-66731, Iran*

**Farshad Saberi-Movahed**  *fmovahed@nvidia.com*
*NVIDIA, Santa Clara, CA 95051, USA*

**Jens Lehmann**  *jens.lehmann@tu-dresden.de*
*Technische Universität Dresden, Dresden, Germany*
*Amazon (work done outside of Amazon), Dresden, Germany*

**Sahar Vahdati**  *sahar.vahdati@tu-dresden.de*
*Nature-Inspired Machine Intelligence Group, ScaDS.AI center, Technische Universität Dresden, Dresden, Germany*
*Institute for Applied Computer Science (InfAI), Dresden, Germany*

**Reviewed on OpenReview:** *https://openreview.net/forum?id=LNvbgBFPMt*

## Abstract

Current methods of feature selection based on a self-representation framework use all the features of the original data in their representation framework. This issue carries over redundant and noisy features into the representation space, thereby diminishing the quality and effectiveness of the results. This work proposes a novel representation learning method, dubbed GRSSLFS (Graph Regularized Self-Representation and Sparse Subspace Learning), that mitigates the drawbacks of using all features. GRSSLFS employs an approach for constructing a basis for the feature space, which includes those features with the highest variance. The objective function of GRSSLFS is then developed based on a self-representation framework that combines subspace learning and matrix factorization of the basis matrix. Moreover, these basis features are incorporated into a manifold learning term to preserve the geometrical structure of the underlying data. We provide an evaluation of effectiveness and performance of GRSSLFS on several widely used benchmark datasets. The results show that GRSSLFS achieves a high level of performance compared to several classic and state-of-the-art feature selection methods.

## 1 Introduction

Feature selection is a prominent technique for managing high-dimensional data, aiming to identify the most representative and effective features within the original feature set Dinh & Ho (2020); Majumdar & Chatterjee (2022). Feature selection is applied across various tasks, including image processing Shi et al.

---

[*]Corresponding author

(2023), bioinformatics Moslemi (2023), and genomics Sun et al. (2022). Moreover, recent advances in this field involves different perspectives such as subspace learning Li et al. (2021), graph learning Roffo et al. (2021) and self-representation learning Chen et al. (2022).

Subspace learning is a concept that aims to find the best lower-dimensional subspace within the original high-dimensional space Ren et al. (2021). These methods can be categorized into two primary types: Linear Subspace Learning, and Nonlinear Subspace Learning. The first category, which comprises well-established techniques such as Local Preserving Projection (LPP) He & Niyogi (2003) and Principal Component Analysis (PCA) Marukatat (2023), focuses on recognizing linear combinations of the initial features that define a lower-dimensional subspace. Nonetheless, in situations where the inherent data structure departs from linearity, nonlinear subspace learning methods become inevitable. Methods like Kernel PCA Ghojogh et al. (2023) and Locally Linear Embedding (LLE) Roweis & Saul (2000) serve as examples of approaches for nonlinear subspace learning. While subspace learning and feature selection can be employed separately, combining them can result in even more potent approaches to dimensionality reduction in data and modeling strategies. For instance, MFFS Wang et al. (2015a) stands out as a prominent example of unsupervised feature selection methods developed within the context of subspace learning and matrix factorization techniques. Subsequently, numerous approaches that integrate subspace learning and feature selection have been introduced, such as those presented in SLSDR Shang et al. (2020), DGSLFS Sheng et al. (2021), CNAFS Yuan et al. (2022) and SPLR Li et al. (2022).

In addition to subspace learning, representation learning has been widely used in various domains of dimensionality reduction methods Shang et al. (2021); Wang et al. (2023). When it comes to deal with high-dimensional data with a large amount of redundant features, it will be a complex task to analyze and mine such data, which will generally lead to imprecise results. One preliminary goal of representation learning is to discover basic information, such as the inherent structure and characteristic, from the data Wu et al. (2023); Shao et al. (2023). As a popular category of representation learning, self-representation learning achieves high performance in combination with other methods such as subspace learning Chen et al. (2022). Self-representation learning indeed originates from one of the earliest mathematical concepts called a basis for vector spaces and is rested on the assumption that every sample or feature vector of the original data can be described in terms of other sample or feature vectors. According to this merit of self-representation, some effective feature selection methods have been recently developed to focus on utilizing the correlations among features Tang et al. (2019); Zhang et al. (2022); Lin et al. (2022).

Feature selection techniques that fall under either the subspace learning or self-representation learning framework are a prevalent category within the realm of dimensionality reduction. However, these techniques encompass all the features from the initial dataset, including redundant ones, in their representations, which can potentially harm the effectiveness of feature selection. In order to get the root of such problem, the philosophy behind the basis for vector spaces, which forms the foundation for the development of self-representation-based feature selection methods, can be revisited. In linear algebra, it is known that a basis for a vector space can be defined as a set of linearly independent vectors that can uniquely produce the whole space. In fact, because of the linear independency among the members of a basis, it is highly probable that redundant vectors are excluded from the basis. Hence, with a basis for the space formed by the features of the original data, redundant features will have a diminished impact on the self-representation process, thereby allowing their exclusion from the set of selected features. The comprehensive properties of the basis for the feature space, namely the linear independence of its members and the unique representation of the feature space, serve as motivation for us to introduce a more efficient form of self-representation and subspace learning frameworks for feature selection.

The aim of this paper is to establish a new type of feature selection method which is called Graph Regularized Self-Representation and Sparse Subspace Learning (GRSSLFS). The primary contributions of the paper are outlined as follows:

- Expressing both the subspace learning and self-representation problems by employing a basis for the feature space. Moreover, GRSSLFS merges the subspace learning and self-representation problems, leveraging the basis of the feature space, to simultaneously eliminate redundant features and perform feature selection.

- To the best of our knowledge, GRSSLFS is the first feature selection method that integrates a basis of linearly independent features into a unified framework of subspace learning and self-representation.

- Introducing the Variance Basis Extension (VBE) framework, which utilizes the variance information of features to establish a basis for the feature space. The objective is to identify basis features with the highest variance scores, ensuring that the basis for the feature space consists of elements with the greatest dispersion.

- Through conducting comprehensive experiments on various real-world datasets, the results indicate that GRSSLFS outperforms several existing related and state-of-the-art feature selection methods. Additionally, the application of GRSSLFS on the PneumoniaMNIST dataset Yang et al. (2021) is assessed to precisely detect and analyze the cardiac silhouette in chest X-ray images.

Finally, the key characteristics of some unsupervised feature selection methods based on subspace learning or self-representation, together with our proposed GRSSLFS are outlined in Table 1. The detailed descriptions of these methods are provided in Appendix A.1.

Table 1: Comparison of some related unsupervised feature selection methods.

| Method | Subspace Learning | Self-representation | Graph Regularization | Sparse Regularization | Orthogonality Constraint |
|---|:---:|:---:|:---:|:---:|:---:|
| **MFFS** Wang et al. (2015a) | ✓ | × | × | × | ✓ |
| **MPMR** Wang et al. (2015b) | ✓ | × | × | × | ✓ |
| **RMFFS** Qi et al. (2018) | ✓ | × | × | ✓ | × |
| **DSRMR** Tang et al. (2018) | × | ✓ | ✓ | ✓ | × |
| **NSSLFS** Zheng et al. (2019) | ✓ | × | × | ✓ | × |
| **SCFS** Parsa et al. (2020) | ✓ | ✓ | × | ✓ | ✓ |
| **RNE** Liu et al. (2020) | ✓ | × | × | × | ✓ |
| **SLSDR** Shang et al. (2020) | ✓ | × | ✓ | ✓ | ✓ |
| **DGSLFS** Sheng et al. (2021) | ✓ | × | ✓ | ✓ | ✓ |
| **CNAFS** Yuan et al. (2022) | ✓ | × | ✓ | ✓ | ✓ |
| **SPLR** Li et al. (2022) | ✓ | × | ✓ | ✓ | ✓ |
| **RDMRS2FS** Chen et al. (2022) | × | ✓ | ✓ | ✓ | × |
| **GRSSLFS (Ours)** | ✓ | ✓ | ✓ | ✓ | × |

## 2 Methodology

In this section, we first introduce our approach to constructing a basis for the feature space and then explain how we use this basis in the frameworks of self-representation and subspace learning to propose the GRSSLFS method. In what follows, $\mathbf{X} \in \mathbb{R}_+^{m \times n}$ indicates the data matrix with $m$ samples and $n$ features. Here, $\mathbb{R}_+^{m \times n}$ denotes the set of non-negative matrices of size $m \times n$. For an $m$ dimension vector space $\mathcal{P}$, the set $B \subseteq \mathcal{P}$ is called a basis if the span of elements of $B$ can produce $W$ and $B$ is linearly independent. Here, the span of some vectors refers to all the possible linear combinations of them. The Frobenius norm, the $L_{2,1}$-norm, the trace, and the transpose of a matrix $\mathbf{A}$ are denoted by $\|\mathbf{A}\|_F$, $\|\mathbf{A}\|_{2,1}$, $\mathrm{Tr}(\mathbf{A})$, and $\mathbf{A}^T$, respectively. Finally, $\langle \mathbf{a}, \mathbf{b} \rangle = \mathbf{a}^T \mathbf{b}$ is the Euclidean inner product of the vectors $\mathbf{a}$ and $\mathbf{b}$.

### 2.1 Constructing a Basis for the Feature Space

A basis for a vector space can be defined as a set of linearly independent vectors that can produce the whole space. This section introduces a novel framework, called Variance Basis Extension (VBE), to select a basis for the space generated by the original features. Let us consider the data matrix $\mathbf{X} = [\mathbf{f}_1, \mathbf{f}_2, \ldots, \mathbf{f}_n]$ so that the rank of $\mathbf{X}$ is $r$. The Basis Extension (BE) method Meyer (2000) starts by choosing an arbitrary feature like $\mathbf{f}_{i_1}$ from among all the features and sets $B = \{\mathbf{f}_{i_1}\}$. It is clear that $B$ is linearly independent. If $r = 1$, the basis building process by the BE method terminates at this point, and $B$ will be a basis. Otherwise, we have $r \leq n$, and we can choose a new feature like $\mathbf{f}_{i_2}$ from $\{\mathbf{f}_1, \mathbf{f}_2, \ldots, \mathbf{f}_n\} - \{\mathbf{f}_{i_1}\}$ so that $B \cup \{\mathbf{f}_{i_2}\} = \{\mathbf{f}_{i_1}, \mathbf{f}_{i_2}\}$ is a linearly independent set. The process described in these two steps continues in the same way, and it can be easily proved that during the $r$ steps, the BE method is able to construct the set $\{\mathbf{f}_{i_1}, \mathbf{f}_{i_2}, \ldots, \mathbf{f}_{i_r}\}$, which is a basis for the space generated by the whole features.

As seen in the first step of the BE method, we are allowed to choose any arbitrary vector from $\{\mathbf{f}_1, \mathbf{f}_2, \ldots, \mathbf{f}_n\}$. Therefore, there is an ideal situation to choose features that contain useful information from the data and can

be effective in the feature selection process. As one of the most popular tools in data mining, the variance information of the data has gained widespread acceptance in some well-known dimensionality reduction methods such as the PCA and variance score methods. The primary motivation for the use of the variance information stems from its simple implementation and its power to display the amount of data dispersion. According to this advantage, the variance score method simply but effectively maintains a number of features with the highest variance score and removes the remaining features.

In the following, we introduce our VBE framework which integrates the variance information of features into the BE method with the aim of constructing a basis of features with the highest variance score. The VBE method involves the following three steps:

- For each feature $\mathbf{f}_i$, $i = 1, 2, \ldots, n$, its variance score is calculated. Then, the set of features $\{\mathbf{f}_1, \mathbf{f}_2, \ldots, \mathbf{f}_n\}$ is sorted in descending order based on their variance score. Next, we will have $\{\mathbf{f}'_1, \mathbf{f}'_2, \ldots, \mathbf{f}'_n\}$ where $\mathbf{f}'_i$ refers to the case that $\mathbf{f}'_i$ has the $i$th-ranked among all features based on the variance scores. Now, we set $\mathbf{X}' = [\mathbf{f}'_1, \mathbf{f}'_2, \ldots, \mathbf{f}'_n]$.

- Set $B = \{\mathbf{f}'_1\}$ so that $\mathbf{f}'_1$ has the highest variance score.

- Let rank$(\mathbf{X}) = r$. For $k = 2, \ldots, n$, if $\mathbf{f}'_k \notin \text{span}(B)$, set $B = B \cup \{\mathbf{f}'_k\}$. In this case, if the number of members of $B$ is equal to $r$, then the VBE method terminates at this point, and $B$ will is a basis for the space generated by the features $\{\mathbf{f}_1, \mathbf{f}_2, \ldots, \mathbf{f}_n\}$.

The main difference between the proposed VBE method and its original BE method is that the features selected by VBE have the highest variance score. This method gives a basis with the members that have the most dispersion in space and avoid the accumulation of the features as much as possible.

## 2.2 Self-representation Problem and the Basis

In the field of linear algebra, a direct connection exists between redundancy and the concept of linear independency. Each basis for the feature space exhibits two essential properties. Firstly, a basis can represent the complete feature space through its elements, achieved with a significantly reduced number of elements compared to the entire feature space. The second characteristic of a basis is that its elements are linearly independent which leads to a significant reduction in data redundancy which is a crucial aim in feature selection. Continuing on, it is discussed how the concept of a basis for the feature space can lead to the definition of an effective form for the self-representation problem.

The self-representation problem of features is considered as a linear representation of all features in a dataset, including redundant ones which can potentially harm the effectiveness of feature representation. Formally, the self-representation problem of features is defined as, $\min_{\mathbf{C} \in \mathbb{R}^{n \times n}} \|\mathbf{X} - \mathbf{X}\mathbf{C}\|_F^2$, which can also be expressed as:

$$\mathbf{X} \simeq \mathbf{X}\mathbf{C} \Rightarrow \begin{cases} \mathbf{f}_1 \simeq c_{11}\mathbf{f}_1 + c_{21}\mathbf{f}_2 + \cdots + c_{n1}\mathbf{f}_n, \\ \mathbf{f}_2 \simeq c_{12}\mathbf{f}_1 + c_{22}\mathbf{f}_2 + \cdots + c_{n2}\mathbf{f}_n, \\ \vdots \\ \mathbf{f}_n \simeq c_{1n}\mathbf{f}_1 + c_{2n}\mathbf{f}_2 + \cdots + c_{nn}\mathbf{f}_n, \end{cases} \tag{1}$$

where $\mathbf{X} = [\mathbf{f}_1, \mathbf{f}_2, \ldots, \mathbf{f}_n]$ and $\mathbf{C} = [c_{ij}]$ is the coefficient matrix. Without loss of generality, let us assume that rank$(\mathbf{X}) = m$, where $m \leq n$. In the case, where $m > n$, a similar discussion can be also applied. Under this assumption, it can be shown that there exists a basis $B$ with linearly independent features $\{\mathbf{f}_{i_1}, \ldots, \mathbf{f}_{i_m}\}$ such that the set $\{\mathbf{f}_1, \mathbf{f}_2, \ldots, \mathbf{f}_n\}$ can be represented by the basis $B$. Therefore, the following representation based on the basis can be obtained for the features:

$$\begin{cases} \mathbf{f}_1 \simeq g_{11}\mathbf{f}_{i_1} + g_{12}\mathbf{f}_{i_2} + \cdots + g_{m1}\mathbf{f}_{i_m}, \\ \mathbf{f}_2 \simeq g_{12}\mathbf{f}_{i_1} + g_{22}\mathbf{f}_{i_2} + \cdots + g_{m2}\mathbf{f}_{i_m}, \\ \vdots \\ \mathbf{f}_n \simeq g_{1n}\mathbf{f}_{i_1} + g_{m2}\mathbf{f}_{i_2} + \cdots + g_{mn}\mathbf{f}_{i_m}, \end{cases} \tag{2}$$

which implies that $\mathbf{X} \simeq \mathbf{BG}$, where $\mathbf{B} = [\mathbf{f}_{i_1}, \ldots, \mathbf{f}_{i_m}] \in \mathbb{R}^{m \times m}$ is a basis matrix, and $\mathbf{G} = [g_{ij}] \in \mathbb{R}^{m \times n}$ is the basis coefficient matrix. Now, compared to the linear system of equations (1), the self-representation problem based on the basis can be defined as:

$$\min_{\mathbf{G}} \|\mathbf{X} - \mathbf{BG}\|_F^2. \tag{3}$$

Problem (3) clarifies that rather than utilizing all features, many of which may be redundant, it is feasible to introduce a new formulation for the self-representation problem, which utilizes a basis containing linearly independent features.

## 2.3 Subspace Learning Problem and the Basis

A popular technique in machine learning is subspace learning, which is used to discover low-dimensional representations of high-dimensional data through linear or non-linear mappings. Several subspace learning models using Euclidean distance have been introduced in the past few years Wang et al. (2015a); Qi et al. (2018); Shang et al. (2020). Among the efficient subspace learning models is the following one:

$$\mathrm{dist}_{\mathrm{SL}}(\mathbf{X}, \mathbf{XU}) = \min_{\mathbf{V}} \|\mathbf{X} - \mathbf{XUV}\|_F^2, \tag{4}$$

where $\mathbf{U} \in \mathbb{R}^{n \times k}$ and $\mathbf{V} \in \mathbb{R}^{k \times n}$ are called the feature weight and the representation matrices, respectively. We shall note that the subspace learning model (4) indeed calculates the distance between the space of features generated by $\mathbf{X}$ and the space of the selected subset of features generated by $\mathbf{XU}$. As a result, if $\mathbf{U}$ is chosen such that $\mathbf{XU} = \mathbf{B}$, then it can be seen that the distance between $\mathbf{X}$ and $\mathbf{B}$ is zero, i.e., $\mathrm{dist}_{\mathrm{SL}}(\mathbf{X}, \mathbf{B}) = 0$, which means that the space of $\mathbf{X}$ and the space of $\mathbf{B}$ can align or be identical. More information regarding this issue is provided in the following theorem.

**Theorem 2.1.** *Let $\mathbf{X} = [\mathbf{f}_1, \ldots, \mathbf{f}_n] \in \mathbb{R}^{m \times n}$ be a dataset with $n$ features such that $\mathrm{rank}(\mathbf{X}) = m$. Let us also assume that $\mathbf{B} = [\mathbf{f}_{i_1}, \ldots, \mathbf{f}_{i_m}]$ is a basis for the space generated by $\mathbf{X}$. Then, according to the distance-based subspace learning problem introduced in (4), the distance between $\mathbf{X}$ and $\mathbf{B}$ is zero, that is $\mathrm{dist}_{\mathrm{SL}}(\mathbf{X}, \mathbf{B}) = 0$.*

*Proof.* If the matrix $\mathbf{U}$ is defined as $\mathbf{U} = [\mathbf{u}_{i_1}, \ldots, \mathbf{u}_{i_m}]$, where $\mathbf{u}_{i_j}$ (for $j = 1, \ldots, m$) is a vector whose $i_j$-element is 1 and other elements are 0, then it can be easily seen that $\mathbf{B} = \mathbf{XU}$. With this assumption, the distance-based subspace learning problem (4) will be of the following form:

$$\mathrm{dist}_{\mathrm{SL}}(\mathbf{X}, \mathbf{XU}) = \mathrm{dist}_{\mathrm{SL}}(\mathbf{X}, \mathbf{B}) = \min_{\mathbf{V}} \|\mathbf{X} - \mathbf{BV}\|_F.$$

It turns out from the above problem that $\mathrm{dist}_{\mathrm{SL}}(\mathbf{X}, \mathbf{B}) \leq \|\mathbf{X} - \mathbf{BV}\|_F$, for each $\mathbf{V}$. On the other hand, since $\mathbf{B}$ is a basis for the space generated by $\mathbf{X}$, there is a basis coefficient matrix $\mathbf{G}$ such that $\mathbf{X} = \mathbf{BG}$. Taking this observation into account and assuming $\mathbf{V} = \mathbf{G}$, it becomes evident that $0 \leq \mathrm{dist}_{\mathrm{SL}}(\mathbf{X}, \mathbf{B}) \leq \|\mathbf{X} - \mathbf{BG}\|_F = 0$, which completes the proof. $\square$

**Remark.** It should be noted that in the case where $\mathrm{rank}(\mathbf{X}) < m$ or $m > n$, it is still possible to create a basic matrix for the feature space, and this will not cause any issues in designing our feature selection framework.

Now, we can state the subspace learning model (4) in terms of the basis matrix $\mathbf{B}$. In order to accomplish this goal, let us consider the representation relation (2). Therefore, the expression $\mathbf{X} - \mathbf{XUV}$ can be shown as $\mathbf{X} - \mathbf{BGUV}$. On the other hand, according to the vector space generation property of the basis, since the distance between $\mathbf{X}$ and $\mathbf{B}$ is zero, it is straightforward to see that $\mathbf{X}$ can be substituted using the basis matrix $\mathbf{B}$. As a result of the preceding discussion, a basis-based subspace learning problem for feature selection can be established as follows:

$$\min_{\mathbf{G}, \mathbf{U}, \mathbf{V}} \|\mathbf{B} - \mathbf{BGUV}\|_F^2, \tag{5}$$

where $\mathbf{G} \in \mathbb{R}^{m \times n}, \mathbf{U} \in \mathbb{R}^{n \times k}$, and $\mathbf{V} \in \mathbb{R}^{k \times m}$, such that $k \leq n$ is the dimension of the selected feature space.

### 2.4 The Importance of the Roles of U, V, and G

The framework defined in (5) can be employed as a bedrock for the feature selection process, in which the feature weight matrix $\mathbf{U}$, the representation matrix $\mathbf{V}$, and the basis coefficient matrix $\mathbf{G}$ play a major role in selecting the underlying features in our proposed method.

• **The Role of U:** First, $\mathbf{U}$ is used as the weight matrix in the feature selection process, that is $\mathbf{X}_{\text{selected}} = \mathbf{BGU} = \sum_{i=1}^{n}(\mathbf{BG})_i \mathbf{u}_{i:}$, where $(\mathbf{BG})_i$ is the $i$-th column of $\mathbf{BG}$, and $\mathbf{u}_{i:}$ is the $i$-th row of $\mathbf{U}$. In fact, the sparser the construction of rows in $\mathbf{U}$, the higher the likelihood of selecting effective features. This description allows the utilization of the $L_{2,1}$-norm to promote row sparsity within the matrix $\mathbf{U}$, which can be demonstrated as $\|\mathbf{U}\|_{2,1}$. The importance of row sparsity in the $L_{2,1}$-norm lies in its effectiveness in managing structured sparsity, where the objective is to choose discriminative features.

• **The Role of V:** Second, let us consider that $\mathbf{B} = [\mathbf{f}_{i_1}, \ldots, \mathbf{f}_{i_m}]$ and $\mathbf{V} = [\mathbf{v}_1, \ldots, \mathbf{v}_m]$. It can be concluded from Problem (5) that $\mathbf{B} \simeq \mathbf{BGUV}$, which means that $[\mathbf{f}_{i_1}, \ldots, \mathbf{f}_{i_m}] \simeq [\mathbf{BGUv}_1, \ldots, \mathbf{BGUv}_m]$. This implies, each feature $\mathbf{f}_{i_l}$ of the basis can be represented by $\mathbf{BGUv}_l$, for $l = 1, \ldots, m$. To be more precise, we can see that $\mathbf{f}_{i_l} \simeq \sum_{r=1}^{k}(\mathbf{BGU})_r \mathbf{v}_{rl}$. Hence, the more sparse the columns of $\mathbf{V}$ are, the fewer members of $\mathbf{BGU}$ are used in the representation of $\mathbf{f}_{i_l}$ of the basis matrix $\mathbf{B}$. As a result, a reduction in redundancy is more likely to occur. In order to reflect the importance of the representation matrix $\mathbf{V}$ in our proposed feature selection process, the idea of inner product regularization can be used. Let us consider the representation matrix $\mathbf{V}$. The Gram matrix of $\mathbf{V}$ can be written as $\mathbf{V}^T\mathbf{V} = [\langle \mathbf{v}_l, \mathbf{v}_p \rangle]$, where $\mathbf{v}_l$ is the $l$-th column of $\mathbf{V}$, for $l = 1, \ldots, m$. Now, it can be easily seen that the expression $\text{Tr}(\mathbf{1}_{m \times m}\mathbf{V}^T\mathbf{V}) - \text{Tr}(\mathbf{V}^T\mathbf{V})$ readily corresponds to the summation of the off-diagonal elements within the Gram matrix $\mathbf{V}^T\mathbf{V}$, that is to say $\text{Tr}(\mathbf{1}_{m \times m}\mathbf{V}^T\mathbf{V}) - \text{Tr}(\mathbf{V}^T\mathbf{V}) = \sum_{l,p=1,l \neq p}^{m} \langle \mathbf{v}_l, \mathbf{v}_p \rangle$. Therefore, by the subsequent problem given as:

$$\min_{\mathbf{V}} \left( \text{Tr}(\mathbf{1}_{m \times m}\mathbf{V}^T\mathbf{V}) - \text{Tr}(\mathbf{V}^T\mathbf{V}) \right), \tag{6}$$

and under the assumption that $\mathbf{V}$, it can be deduced that the optimal solution to Problem (6) is attained when the inner product terms $\langle \mathbf{v}_l, \mathbf{v}_p \rangle$ tend toward zero for all pairs of $l$ and $p$ except when $l = p$. On the other hand, since $\langle \mathbf{v}_l, \mathbf{v}_p \rangle = (\mathbf{v}_l)^T\mathbf{v}_p$ and considering the non-negativity assumption for $\mathbf{v}_l$ and $\mathbf{v}_p$, the value of $\langle \mathbf{v}_l, \mathbf{v}_p \rangle$ tends towards zero when either $\mathbf{v}_l$ or $\mathbf{v}_p$ become zero or extremely sparse vectors. Consequently, the optimization Problem (6) can result in a significant degree of sparsity within the coefficient matrix $\mathbf{V}$.

Considering the importance of the role of $\mathbf{U}$ and $\mathbf{V}$, we integrate two efficient sparsity regularizations into the objective function (5) in order to make both $\mathbf{U}$ and $\mathbf{V}$ sparse. Two of the most useful and simple tools are the $L_{2,1}$-norm and the inner product regularizations, which benefit from the robustness in a row and column sparsity, respectively. We can now state the sparse form of the subspace learning problem based on the basis as follows:

$$\min_{\mathbf{G},\mathbf{U},\mathbf{V}} \left( \|\mathbf{B} - \mathbf{BGUV}\|_F^2 + \|\mathbf{U}\|_{2,1} + \left( \text{Tr}(\mathbf{1}_{m \times m}\mathbf{V}^T\mathbf{V}) - \text{Tr}(\mathbf{V}^T\mathbf{V}) \right) \right).$$

• **The Role of G:** Graph regularization is essential in feature selection methods as it integrates the inherent relationships between features into the selection process Khoshraftar & An (2024). In doing so, graph regularization aids in choosing features not solely based on their individual predictive capabilities but also on their relationships with other features. Typically, graph regularization is expressed through an additional term in the objective function of feature selection algorithms. This term imposes penalties on solutions that deviate from the smoothness enforced by the structure of graph. Common graph regularization terms include the graph Laplacian, which encourages solutions where neighboring nodes in the graph have similar function values Hamilton (2020).

To establish a framework for the graph regularization in our proposed method, let us consider the relation $\mathbf{X} \simeq \mathbf{BG}$ and assume that $\mathbf{G} = [\mathbf{g}_1, \ldots, \mathbf{g}_n]$. It is concluded that $[\mathbf{f}_1, \mathbf{f}_2, \ldots, \mathbf{f}_n] \simeq [\mathbf{Bg}_1, \ldots, \mathbf{Bg}_n]$. As a result, according to the basic principle of preserving the geometric structure in the feature manifold, if the two features $\mathbf{f}_l$ and $\mathbf{f}_r$ have a similar structure in the feature space, then it can be expected that their corresponding representations $\mathbf{Bg}_l$ and $\mathbf{Bg}_r$ will also have the similar structure. This observation can be

expressed as follows:

$$\min_{\mathbf{G}} \mathrm{Tr}\left(\mathbf{BGLG}^T\mathbf{B}^T\right) = \frac{1}{2}\sum_{q,r=1}^{n} a_{qr}\|\mathbf{Bg}_q - \mathbf{Bg}_r\|_2^2, \tag{7}$$

where $\mathbf{A} = [a_{qr}] \in \mathbb{R}^{n\times n}$ is the similarity matrix for the features, and the Laplacian matrix $\mathbf{L}$ is defined as $\mathbf{L} = \mathbf{P} - \mathbf{A}$ such that $\mathbf{P} = \mathrm{diag}(p_{qq})$ is a diagonal matrix with the diagonal entries $p_{qq} = \sum_{r=1}^{n} a_{qr}$, for $q = 1, \ldots, n$. Here, we consider $a_{qr} = \exp\left(\frac{-\|\mathbf{f}_q-\mathbf{f}_r\|_2^2}{t^2}\right)$, if $\mathbf{f}_q \in N_k(\mathbf{f}_r)$ or $\mathbf{f}_r \in N_k(\mathbf{f}_q)$, where $N_k(\mathbf{f}_r)$ refers to the set of $k$ feature vectors that are closest to $\mathbf{f}_r$, and $t$ denotes the scale parameter of the Gaussian distribution, and $a_{qr} = 0$ otherwise. Now, according to (7), the graph regularized self-representation problem based on the basis can be formulated as follows:

$$\min_{\mathbf{G}}\left(\|\mathbf{X} - \mathbf{BG}\|_F^2 + \mathrm{Tr}\left(\mathbf{BGLG}^T\mathbf{B}^T\right)\right). \tag{8}$$

## 2.5 The GRSSLFS Method

Based on the the previous sections, we establish the novel GRSSLFS method according to the combination of "self-representation based on the basis" and "sparse subspace learning based on the basis" as follows:

$$\min_{\mathbf{G},\mathbf{U},\mathbf{V}}\left(\|\mathbf{X} - \mathbf{BG}\|_F^2 + \|\mathbf{B} - \mathbf{BGUV}\|_F^2 + \alpha\mathrm{Tr}\left(\mathbf{BGLG}^T\mathbf{B}^T\right) + \beta\|\mathbf{U}\|_{2,1} + \frac{\gamma}{2}\left(\mathrm{Tr}(\mathbf{1}_{m\times m}\mathbf{V}^T\mathbf{V}) - \mathrm{Tr}(\mathbf{V}^T\mathbf{V})\right)\right), \tag{9}$$

where $\alpha, \beta$ and $\gamma$ are the regularization parameters. Moreover, we assume that $\mathbf{G} \in \mathbb{R}_+^{m\times n}, \mathbf{U} \in \mathbb{R}_+^{n\times k}$, and $\mathbf{V} \in \mathbb{R}_+^{k\times m}$, such that $k \leq n$ is the dimension of the selected feature space. To solve Problem (9), let us consider the function $J(\mathbf{G}, \mathbf{U}, \mathbf{V})$ as follows:

$$J(\mathbf{G}, \mathbf{U}, \mathbf{V}) = \|\mathbf{X} - \mathbf{BG}\|_F^2 + \|\mathbf{B} - \mathbf{BGUV}\|_F^2 + \alpha\mathrm{Tr}\left(\mathbf{BGLG}^T\mathbf{B}^T\right) + \beta\mathrm{Tr}(\mathbf{U}^T\mathbf{EU})$$
$$+ \gamma\left(\mathrm{Tr}(\mathbf{1}_{m\times m}\mathbf{V}^T\mathbf{V}) - \mathrm{Tr}(\mathbf{V}^T\mathbf{V})\right) + \mathrm{Tr}(\mathbf{MG}^T) + \mathrm{Tr}(\mathbf{NU}^T) + \mathrm{Tr}(\mathbf{WV}^T),$$

where $\mathbf{E} \in \mathbb{R}_+^{n\times n}$ is a diagonal matrix with the diagonal elements $\mathbf{E}_{ii} = 1/2\|\mathbf{U}_i\|_2$, for $i = 1, \ldots, n$. Moreover, $\mathbf{M} \in \mathbb{R}^{m\times n}, \mathbf{N} \in \mathbb{R}^{n\times n}$ and $\mathbf{W} \in \mathbb{R}^{k\times n}$ are the Lagrange multipliers. The problem mentioned above can be effectively solved by applying an alternative iterative procedure involving $\mathbf{G}$, $\mathbf{U}$, and $\mathbf{V}$. To achieve this, one variable needs to be fixed, whereas the other variables need to be determined. That is to say

• **Update G with fixed U and V.** Taking the partial derivative of $J$ in terms of $\mathbf{G}$ shows

$$\frac{\partial J}{\partial \mathbf{G}} = -2\mathbf{B}^T\mathbf{X} + 2\mathbf{B}^T\mathbf{BG} + 2\alpha\mathbf{B}^T\mathbf{BGL} - 2\mathbf{B}^T\mathbf{BV}^T\mathbf{U}^T + 2\mathbf{B}^T\mathbf{BGUVV}^T\mathbf{U}^T + \mathbf{M}.$$

• **Update U with fixed G and V.** Taking the partial derivative of $J$ in terms of $\mathbf{U}$ shows

$$\frac{\partial J}{\partial \mathbf{U}} = -2\mathbf{G}^T\mathbf{B}^T\mathbf{BV}^T + 2\mathbf{G}^T\mathbf{B}^T\mathbf{BGUVV}^T + 2\beta\mathbf{EU} + \mathbf{N}.$$

• **Update V with fixed G and U.** Finally, taking the partial derivative of $J$ in terms of $\mathbf{V}$ shows

$$\frac{\partial J}{\partial \mathbf{V}} = -2\mathbf{U}^T\mathbf{G}^T\mathbf{B}^T\mathbf{B} + 2\mathbf{U}^T\mathbf{G}^T\mathbf{B}^T\mathbf{BGUV} + 2\gamma(\mathbf{V}\mathbf{1}_{m\times m} - \mathbf{V}) + \mathbf{W}.$$

Now, putting the Karush-Kuhn-Tucker conditions Lee & Seung (2001) together with $\partial J/\partial\mathbf{G} = \partial J/\partial\mathbf{U} = \partial J/\partial\mathbf{V} = 0$ results in the following updating rule:

$$\mathbf{G}_{ij} \leftarrow \mathbf{G}_{ij}\sqrt{\frac{(\mathbf{B}^T\mathbf{X} + \alpha\mathbf{B}^T\mathbf{BGA} + \mathbf{B}^T\mathbf{BV}^T\mathbf{U}^T)_{ij}}{(\mathbf{B}^T\mathbf{BG} + \alpha\mathbf{B}^T\mathbf{BGP} + \mathbf{B}^T\mathbf{BGUVV}^T\mathbf{U}^T)_{ij}}}, \tag{10}$$

$$\mathbf{U}_{ij} \leftarrow \mathbf{U}_{ij} \sqrt{\frac{(\mathbf{G}^T\mathbf{B}^T\mathbf{B}\mathbf{V}^T)_{ij}}{(\mathbf{G}^T\mathbf{B}^T\mathbf{B}\mathbf{G}\mathbf{U}\mathbf{V}\mathbf{V}^T + \beta\mathbf{E}\mathbf{U})_{ij}}}, \tag{11}$$

$$\mathbf{V}_{ij} \leftarrow \mathbf{V}_{ij} \sqrt{\frac{(\mathbf{U}^T\mathbf{G}^T\mathbf{B}^T\mathbf{B} + \gamma\mathbf{V})_{ij}}{(\mathbf{U}^T\mathbf{G}^T\mathbf{B}^T\mathbf{B}\mathbf{G}\mathbf{U}\mathbf{V} + \gamma\mathbf{V}\mathbf{1}_{m\times m})_{ij}}}. \tag{12}$$

**Remark.** Regarding the square root operation applied in the updating rules (10), (11), and (12), it is essential to highlight that the objective function $J$ incorporates the second-order matrix polynomials concerning the variables $\mathbf{G}$, $\mathbf{U}$, and $\mathbf{V}$. To derive efficient update procedures for solving these variables, prior studies Ding et al. (2005); Ding & et al. (2006) suggest the utilization of update rules (like the ones given in (10), (11), and (12) that are founded on the concept of the square root. Algorithm 1 is a summary of the procedure developed above to solve the objective function of GRSSLFS. Moreover, the framework of the proposed GRSSLFS method is displayed in Figure 1.

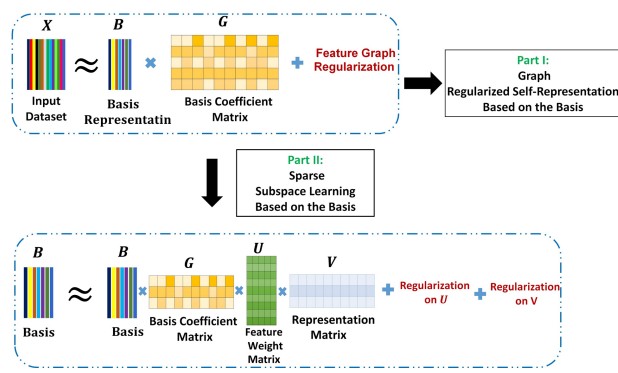

Figure 1: Framework of the proposed GRSSLFS method which integrates the self-representation based on the basis with the sparse subspace learning based on the basis.

**Algorithm 1** The proposed GRSSLFS method.

1: **Input:** Data matrix $\mathbf{X}$; the number of selected features $k$.
2: Construct the basis matrix $\mathbf{B}$ using the VBE method discussed in Section 2.1.
3: Initialize the matrices $\mathbf{G}$, $\mathbf{U}$, and $\mathbf{V}$.
4: **while** not converged **do steps 4 and 6**:
5: Update $\mathbf{G}$ according to the rule (10).
6: Update $\mathbf{U}$ according to the rule (11).
7: Update $\mathbf{V}$ according to the rule (12).
8: **Output:** Put the rows of $\mathbf{U}$ in descending order of value according to the 2-norm. Next, sort the features of $\mathbf{X}$ so that the $k$ features have the highest 2-norm score in $\mathbf{U}$.

### 2.6 Convergence Analysis

In the following, a detailed analysis of the monotone property of the objective function for the GRSSLFS method is conducted.

**Theorem 2.2.** *The objective function of the GRSSLFS method monotonously decreases according to the update rules (10), (11), and (12).*

**Proof.** The proof starts by assuming that the matrices $\mathbf{U} \geq 0$ and $\mathbf{V} \geq 0$ are fixed, and it is shown that the objective function of GRSSLFS monotonously decreases in terms of the other variable $\mathbf{G} \geq 0$. With this assumption, we define the function $Z(\mathbf{G}) = \|\mathbf{X} - \mathbf{B}\mathbf{G}\|_F^2 + \alpha\mathrm{Tr}\left(\mathbf{B}\mathbf{G}\mathbf{L}\mathbf{G}^T\mathbf{B}^T\right) + \|\mathbf{B} - \mathbf{B}\mathbf{G}\mathbf{U}\mathbf{V}\|_F^2$. Using simple calculations, the following can be deduced:

$$Z(\mathbf{G}) = \mathrm{Tr}(\mathbf{X}^T\mathbf{X}) - 2\mathrm{Tr}(\mathbf{X}^T\mathbf{B}\mathbf{G}) + \mathrm{Tr}(\mathbf{G}^T\mathbf{B}^T\mathbf{B}\mathbf{G}) + \alpha\mathrm{Tr}\left(\mathbf{B}\mathbf{G}(\mathbf{P} - \mathbf{A})\mathbf{G}^T\mathbf{B}^T\right) + \mathrm{Tr}(\mathbf{B}^T\mathbf{B})$$
$$- 2\mathrm{Tr}(\mathbf{B}^T\mathbf{B}\mathbf{G}\mathbf{U}\mathbf{V}) + \mathrm{Tr}(\mathbf{V}^T\mathbf{U}^T\mathbf{G}^T\mathbf{B}^T\mathbf{B}\mathbf{G}\mathbf{U}\mathbf{V}).$$

Additionally, suppose that $\mathbf{G}' \geq 0$ is given. It can be easily shown that

$$\mathrm{Tr}(\mathbf{G}^T\mathbf{M}\mathbf{G}\mathbf{N}) \leq \sum_{ij} \frac{(\mathbf{M}\mathbf{G}'\mathbf{N})_{ij}\mathbf{G}_{ij}^2}{\mathbf{G}'_{ij}}, \tag{13}$$

for any matrices $\mathbf{M} \in \mathbb{R}^{m\times m}$ and $\mathbf{N} \in \mathbb{R}^{n\times n}$. Using the inequality (13) in conjunction with the inequality $c > 1 + \log(c)$, for any $c > 0$, we have

$$\bullet -\mathrm{Tr}(\mathbf{X}^T\mathbf{B}\mathbf{G}) \leq -\sum_{l,r}\left((\mathbf{B}^T\mathbf{X})_{lr}\mathbf{G}'_{lr}\left(1 + \log\frac{\mathbf{G}_{lr}}{\mathbf{G}'_{lr}}\right)\right), \tag{14}$$

$$\bullet \operatorname{Tr}(\mathbf{G}^T \mathbf{B}^T \mathbf{B}\mathbf{G}) \le \sum_{l,r} (\mathbf{B}^T \mathbf{B}\mathbf{G}')_{lr} \frac{\mathbf{G}_{lr}^2}{\mathbf{G}'_{lr}}, \tag{15}$$

$$\bullet \operatorname{Tr}(\mathbf{B}\mathbf{G}\mathbf{P}\mathbf{G}^T \mathbf{B}^T) \le \sum_{l,r} (\mathbf{B}^T \mathbf{B}\mathbf{G}'\mathbf{P})_{lr} \frac{\mathbf{G}_{lr}^2}{\mathbf{G}'_{lr}}, \tag{16}$$

$$\bullet -\operatorname{Tr}(\mathbf{B}\mathbf{G}\mathbf{A}\mathbf{G}^T \mathbf{B}^T) \le -\sum_{l,r,q} \left( (\mathbf{B}^T \mathbf{B}\mathbf{G}'\mathbf{A})_{ql} \mathbf{G}'_{lr} \left( 1 + \log \frac{\mathbf{G}_{ql}\mathbf{G}_{lr}}{\mathbf{G}'_{ql}\mathbf{G}'_{lr}} \right) \right), \tag{17}$$

$$\bullet -\operatorname{Tr}(\mathbf{B}^T \mathbf{B}\mathbf{G}\mathbf{U}\mathbf{V}) \le -\sum_{l,r} \left( (\mathbf{B}^T \mathbf{B}\mathbf{V}^T \mathbf{U}^T)_{lr} \mathbf{G}'_{lr} \left( 1 + \log \frac{\mathbf{G}_{lr}}{\mathbf{G}'_{lr}} \right) \right), \tag{18}$$

$$\bullet \operatorname{Tr}(\mathbf{V}^T \mathbf{U}^T \mathbf{G}^T \mathbf{B}^T \mathbf{B}\mathbf{G}\mathbf{U}\mathbf{V}) \le \sum_{l,r} (\mathbf{B}^T \mathbf{B}\mathbf{G}'\mathbf{U}\mathbf{V}\mathbf{V}^T \mathbf{U}^T)_{lr} \frac{\mathbf{G}_{lr}^2}{\mathbf{G}'_{lr}}). \tag{19}$$

Let us now define the following function:

$$\Omega(\mathbf{G}, \mathbf{G}') = \operatorname{Tr}(\mathbf{X}^T \mathbf{X}) - 2 \sum_{l,r} \left( (\mathbf{B}^T \mathbf{X})_{lr} \mathbf{G}'_{lr} \left( 1 + \log \frac{\mathbf{G}_{lr}}{\mathbf{G}'_{lr}} \right) \right) + \sum_{l,r} (\mathbf{B}^T \mathbf{B}\mathbf{G}')_{lr} \frac{\mathbf{G}_{lr}^2}{\mathbf{G}'_{lr}}$$

$$+ \alpha \sum_{l,r} (\mathbf{B}^T \mathbf{B}\mathbf{G}'\mathbf{P})_{lr} \frac{\mathbf{G}_{lr}^2}{\mathbf{G}'_{lr}} - \alpha \sum_{l,r,q} \left( (\mathbf{B}^T \mathbf{B}\mathbf{G}'\mathbf{A})_{ql} \mathbf{G}'_{lr} \left( 1 + \log \frac{\mathbf{G}_{ql}\mathbf{G}_{lr}}{\mathbf{G}'_{ql}\mathbf{G}'_{lr}} \right) \right)$$

$$+ \operatorname{Tr}(\mathbf{B}^T \mathbf{B}) - 2 \sum_{l,r} \left( (\mathbf{B}^T \mathbf{B}\mathbf{V}^T \mathbf{U}^T)_{lr} \mathbf{G}'_{lr} \left( 1 + \log \frac{\mathbf{G}_{lr}}{\mathbf{G}'_{lr}} \right) \right) + \sum_{l,r} (\mathbf{B}^T \mathbf{B}\mathbf{G}'\mathbf{U}\mathbf{V}\mathbf{V}^T \mathbf{U}^T)_{lr} \frac{\mathbf{G}_{lr}^2}{\mathbf{G}'_{lr}}.$$

Now, in view of the relations (14)-(19), it turns out that (1) $\Omega(\mathbf{G}, \mathbf{G}) = Z(\mathbf{G})$ and (2) $Z(\mathbf{G}) \le \Omega(\mathbf{G}, \mathbf{G}')$, for any $\mathbf{G} \ge 0$. Taking these observations into account, it can be inferred that $\Omega(\mathbf{G}, \mathbf{G})$ represents an auxiliary function for $Z(\mathbf{G})$ Lee & Seung (2001). Consequently, using the following relation

$$\mathbf{G}^* = \arg \min_{\mathbf{G}} \Omega(\mathbf{G}, \mathbf{G}'), \tag{20}$$

the objective function of the GRSSLFS method monotonously decreases in terms of the variable $\mathbf{G}$. In order to compute the minimization problem (20), the derivative of $\Omega(\mathbf{G}, \mathbf{G}')$ in terms of $\mathbf{G}_{ij}$ leads us to the following relation:

$$\frac{\partial \Omega(\mathbf{G}, \mathbf{G}')}{\partial \mathbf{G}_{ij}} = -2(\mathbf{B}^T \mathbf{X})_{ij} \frac{\mathbf{G}'_{ij}}{\mathbf{G}_{ij}} + 2(\mathbf{B}^T \mathbf{B}\mathbf{G}')_{ij} \frac{\mathbf{G}_{ij}}{\mathbf{G}'_{ij}} + 2\alpha(\mathbf{B}^T \mathbf{B}\mathbf{G}'\mathbf{P})_{ij} \frac{\mathbf{G}_{ij}}{\mathbf{G}'_{ij}} - 2\alpha(\mathbf{B}^T \mathbf{B}\mathbf{G}'\mathbf{A})_{ij} \frac{\mathbf{G}'_{ij}}{\mathbf{G}_{ij}}$$

$$- 2(\mathbf{B}^T \mathbf{B}\mathbf{V}^T \mathbf{U}^T)_{ij} \frac{\mathbf{G}'_{ij}}{\mathbf{G}_{ij}} + 2(\mathbf{B}^T \mathbf{B}\mathbf{G}'\mathbf{U}\mathbf{V}\mathbf{V}^T \mathbf{U}^T)_{ij} \frac{\mathbf{G}_{ij}}{\mathbf{G}'_{ij}}.$$

Thus, assuming that $\partial \Omega(\mathbf{G}, \mathbf{G}')/\partial \mathbf{G}_{ij} = 0$, it follows that

$$\left( \mathbf{B}^T \mathbf{X} + \alpha \mathbf{B}^T \mathbf{B}\mathbf{G}'\mathbf{A} + \mathbf{B}^T \mathbf{B}\mathbf{V}^T \mathbf{U}^T \right)_{ij} (\mathbf{G}'_{ij})^2 = \left( \mathbf{B}^T \mathbf{B}\mathbf{G}' + \alpha \mathbf{B}^T \mathbf{B}\mathbf{G}'\mathbf{P} + \mathbf{B}^T \mathbf{B}\mathbf{G}'\mathbf{U}\mathbf{V}\mathbf{V}^T \mathbf{U}^T \right)_{ij} \mathbf{G}_{ij}^2,$$

which leads to the following relation:

$$\mathbf{G}_{ij} \leftarrow \mathbf{G}'_{ij} \sqrt{\frac{(\mathbf{B}^T \mathbf{X} + \alpha \mathbf{B}^T \mathbf{B}\mathbf{G}'\mathbf{A} + \mathbf{B}^T \mathbf{B}\mathbf{V}^T \mathbf{U}^T)_{ij}}{(\mathbf{B}^T \mathbf{B}\mathbf{G}' + \alpha \mathbf{B}^T \mathbf{B}\mathbf{G}'\mathbf{P} + \mathbf{B}^T \mathbf{B}\mathbf{G}'\mathbf{U}\mathbf{V}\mathbf{V}^T \mathbf{U}^T)_{ij}}}. \tag{21}$$

In conclusion, solving the minimization problem (20) leads to a solution in the form shown in (21), which is exactly consistent with the relation (10) introduced as the update rule for variable $\mathbf{G}$. Here, the proof for the monotone property of the objective function of the GRSSLFS method in terms of $\mathbf{G}$ is completed. Similar to what has been explained above for the case of $\mathbf{G}$, it is possible to prove the monotone property of the objective function of GRSSLFS in terms of $\mathbf{U}$ and $\mathbf{V}$, which are omitted due to the similarity in the process of existing relations.

## 2.7 Complexity Analysis

Let $m, n$ and $k$ be the number of samples, the number of features and the number of selected features, respectively. In order to update the matrices $\mathbf{G}, \mathbf{U}$, and $\mathbf{V}$, several matrix multiplications must be done in GRSSLFS. From the updating rules (10), (11), and (12), it can be found that among all the operations, the most time-consuming parts to update of $\mathbf{G}$, $\mathbf{U}$ and $\mathbf{V}$ are $\mathbf{B}^T\mathbf{B}\mathbf{G}\mathbf{U}\mathbf{V}\mathbf{V}^T\mathbf{U}^T$, $\mathbf{G}^T\mathbf{B}^T\mathbf{B}\mathbf{G}\mathbf{U}\mathbf{V}\mathbf{V}^T$, and $\mathbf{U}^T\mathbf{G}^T\mathbf{B}^T\mathbf{B}\mathbf{G}\mathbf{U}\mathbf{V}$, respectively, so that the time complexity of each of them is almost equal to $O(k^2m^2n^2)$ per iteration. Besides, the required time complexity is $O(n^2)$ for constructing the feature graph. In summary, assuming that $k \leq \{m, n\}$, the computational complexity of GRSSLFS is almost equal to $O(m^2n^2)$.

# 3 Experimental Studies

In this section, we conduct empirical comparisons between our proposed approach, GRSSLFS, and a range of other supervised and unsupervised feature selection techniques. The unsupervised methods consist of LS He et al. (2005), CNAFS Yuan et al. (2022), OCLSP Lin et al. (2022), RNE Liu et al. (2020), VCSDFS Karami et al. (2023), and CAE Balın et al. (2019). Furthermore, the supervised methods examined include Laaso Tibshirani (1996), SLAP Chen et al. (2018a), and CD-LSR Xu et al. (2023). We evaluate the performance of the comparison algorithms in terms of how the selected features impact the downstream task of clustering in various datasets. We refer to Appendix A.2 for the detailed information on benchmark datasets, Appendix A.3 for the compared feature selection algorithms, Appendix A.4 for the evaluation metrics, and Appendix A.5 for the experimental settings details.

Table 2: Comparing the best values of the clustering ACC and NMI of ten algorithms on all datasets. The values in parentheses denote the selected number of features at which the ACC and NMI values are reported.

| ACC | | | | | | | | | | |
|---|---|---|---|---|---|---|---|---|---|---|
| Datasets | Lasso | LS | CNAFS | OCLSP | RNE | SLAP | VCSDFS | CAE | CD-LSR | GRSSLFS |
| CNS | 60.33 ± 1.45(40) | 58.66 ± 1.02 (80) | 63.84 ± 1.62 (50) | 65.31 ± 1.94 (70) | 58.33 ± 0.00 (40) | 63.86 ± 1.17 (70) | 66.49 ± 2.11 (20) | 66.61±0.05(60) | 61.67±1.83(60) | **73.33 ± 0.01 (20)** |
| GLIOMA | **62.20 ± 1.01(40)** | 44.00 ± 0.00 (10) | 50.11 ± 2.83 (40) | 50.77 ± 1.03 (40) | 47.11 ± 2.19 (20) | 48.71 ± 2.01 (30) | 51.23 ± 0.77 (30) | 54.52±3.13(40) | 48.70±3.92(10) | 54.10 ± 2.71(30) |
| TOX171 | 43.71 ± 0.72(30) | 40.35 ± 0.72 (10) | 48.06 ± 1.32 (90) | 48.67 ± 0.89 (40) | 51.69 ± 0.48 (40) | 44.14 ± 0.78 (80) | 52.69 ± 1.26(40) | 53.36±1.24(40) | 48.77±4.18(10) | **57.39 ± 1.27 (40)** |
| SRBCT | 49.58 ± 2.82 (40) | 46.02 ± 2.96 (100) | 44.17 ± 1.55 (60) | 44.27 ± 2.07 (60) | 42.12 ± 3.37 (80) | 43.16 ± 2.34 (90) | 44.39 ± 0.91 (30) | 46.08±4.18(90) | 52.63±4.39(10) | **53.49 ± 3.11 (60)** |
| SMK | 57.22 ± 0.56(10) | 55.11± 2.35 (80) | 60.12 ± 1.72 (40) | 62.31 ± 3.00 (100) | 61.51 ± 0.34 (10) | 62.01± 3.44 (100) | 62.91 ± 2.62 (40) | 61.80±2.06(40) | 64.71±2.72(30) | **65.72 ± 2.59 (10)** |
| ATT | 56.31 ± 4.03(100) | 46.75 ±3.87(60) | 58.68 ±2.78(100) | 59.71 ±1.94(100) | 59.08 ± 3.12(50) | 59.96±2.81(100) | 61.97 ± 1.83 (100) | 56.86±2.79(90) | 58.03±3.33(100) | **64.11 6± 2.65(100)** |
| ORL | 48.95 ± 2.91(100) | 38.51 ± 1.66(90) | 49.17±2.52(90) | 50.45 ±3.41(100) | 51.31 ±3.01(100) | 48.92±1.23(100) | 47.51±1.38(100) | 50.09±2.42(100) | **54.05±3.51(100)** | 53.45 ± 3.13(80) |
| warpAR10P | 39.46 ± 2.18(70) | 21.50 ±0.94(10) | 38.83 ±2.28(10) | 41.68±3.31(10) | 34.62 ±2.45(10) | 39.81±2.19(10) | 36.85 ±0.73(10) | 39.26±2.83(10) | 31.94±2.78(30) | **46.66 ± 3.51(10)** |

| NMI | | | | | | | | | | |
|---|---|---|---|---|---|---|---|---|---|---|
| Datasets | Lasso | LS | CNAFS | OCLSP | RNE | SLAP | VCSDFS | CAE | CD-LSR | GRSSLFS |
| CNS | 7.00 ± 0.11(50) | 2.44 ± 0.45 (90) | 8.15 ± 3.09 (70) | 10.76 ± 3.68 (70) | 2.29 ± 0.00 (50) | 12.76 ± 1.41 (50) | 10.35 ± 2.62 (50) | 11.05±0.41(30) | 13.76±0.06(20) | **18.56 ± 0.00 (70)** |
| GLIOMA | **53.45± 2.16(50)** | 18.67± 0.29 (50) | 27.21 ± 2.43 (20) | 28.93 ± 1.86 (30) | 25.84 ± 4.20 (20) | 27.73± 0.85 (30) | 28.92 ± 1.29 (40) | **32.83±1.83(40)** | 25.35±1.68(70) | 32.09 ± 2.14 (40) |
| TOX171 | 17.99± 1.03(30) | 11.68± 1.97 (90) | 23.47 ± 1.98(80) | 25.27 ± 2.31 (80) | 26.78 ± 0.08 (40) | 28.28± 1.07 (80) | 30.98 ± 0.77 (100) | 31.12±0.31(40) | 31.28±0.93(10) | **36.96 ± 0.60 (100)** |
| SRBCT | 40.88 ± 2.37(20) | 56.23± 4.06 (60) | 41.21 ± 4.36 (60) | 42.44 ± 2.68 (90) | 29.48 ± 3.52 (80) | 41.17± 2.36 (90) | 48.00 ± 4.01 (90) | 45.28±3.59(80) | 49.53±3.86(30) | **59.67 ± 4.18 (90)** |
| SMK | 5.35 ± 0.08(30) | 2.17± 0.05 (80) | 3.75 ± 0.09 (40) | 4.72 ± 0.22 (30) | 3.77 ± 0.08 (10) | 3.53± 0.12 (40) | 5.63 ± 1.55 (40) | 3.51±0.09(80) | 7.12±1.05(30) | **8.20 ± 0.53 (10)** |
| ATT | 71.21±0.89(100) | 68.46 ± 1.37(70) | 77.05± 1.44(90) | 78.61 ±2.53(100) | 77.17 ± 1.04(80) | 79.58±3.98(100) | 78.62 ±0.81(100) | 75.27±2.24(90) | 77.02±2.19(100) | **81.71± 1.31(100)** |
| ORL | 58.23 ±2.13(100) | 63.85 ±1.23(100) | 71.44 ±1.16(100) | 72.19 ±2.34(100) | 72.59±1.33(100) | 72.41 ±2.41(100) | 73.02 ±1.49(100) | 73.64±1.41(100) | 72.57±2.02(100) | **74.56 ± 1.72(80)** |
| warpAR10P | 42.45 ± 1.98(80) | 19.39 ±1.84(40) | 43.00 ±1.64(80) | 45.46 ±3.08(20) | 35.85±4.38(20) | 47.72±1.84(20) | 41.62 ±1.22(80) | 42.35±2.23(80) | 34.22±3.37(20) | **48.63 ± 2.83(10)** |

## 3.1 Results and Analysis

The best value of ACC and NMI metrics at a specific number of selected features is listed in Table 2. It is evident from these results (please see Appendix A.6 for more illustration) that our proposed method, GRSSLFS, outperforms all other nine methods in terms of selecting features that result in better clustering performance. There are only two exceptions (on the GLIOMA and ORL datasets) where the Lasso, CAE and CD-LSR methods led to a higher ACC or NMI compared to our method. For example, the ACC corresponding to Lasso is 62.2 versus 54.10 in our method. Despite the superiority of our method over other methods, its behavior across all datasets is not the same. For example, we can refer to the CNS dataset, where the NMI values are considerably lower compared to the

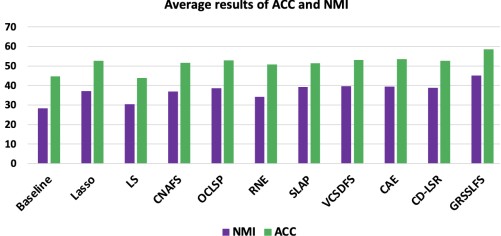

Figure 2: Average of the best ACC and NMI results over all datasets for each feature selection method.

ACC values. This trend is also observed for the other nine feature selection methods. It is mainly due to

the nature of datasets, some of which pose difficulties in selecting distinguishable features and hence have a higher clustering accuracy.

On average, the results of the VCSDFS, CAE and CD-LSR methods are closer to those of the proposed method, compared to other methods. Nevertheless, the proposed method is still superior to such methods for two main reasons. First, the basis vectors in the GRSSLFS method are constructed as an independent linear combination of the selected features. Second, the GRSSLFS method uses self-representation learning along with subspace learning in which the basis of features is used to build both the self-representation and subspace feature spaces. This combination of self-representation and subspace learning with the underlying basis of features makes the proposed method efficient compared to other methods. The effectiveness is observable in Table 2 where the consistency of our method compared to the other methods across all the different datasets is evident. For instance, the ACC values in Table 2 show that while Lasso has a superior performance with respect to GRSSLFS on the GLIOMA dataset, its performance degrades on the other datasets. To summarize the final clustering performance, the average value of ACC and NMI across all the datasets for each feature selection algorithm is also demonstrated as a bar plot in Figure 2. On this account, Figure 2 confirm the superiority of the proposed feature selection model in almost all cases (please refer to the Appendix section for Clustering Performance Results in Figures 4 and 5, Running Time Comparison in A.8, Convergence Test in A.9, and Non-parametric Statistical Test in A.10). In particular, comparing our proposed GRSSLFS method (an unsupervised approach) with other supervised methods, such as Lasso, SLAP, and CD-LSR, reveals that our unsupervised feature selection technique is fully capable of competing with these supervised alternatives.

## 3.2 The Role of Hyperparameters

In this part, the effect of hyperparameters on the performance of our proposed method is investigated. To achieve this, we conduct an ablation study and sensitivity analysis. The objective function of GRSSLFS in (9) has three adjustable parameters; $\alpha$ is for the feature graph regularization term in the self-representation framework and $\beta$ and $\gamma$ control the sparsity of the feature weight matrix $\mathbf{U}$ and representation matrix $\mathbf{V}$, respectively, in the subspace learning process.

For the ablation study, we isolate the impact of the hyperparameters by setting their values to zero, as shown in the configurations in the first column of Table 3. We then choose one biological (GLIOMA) and one image (warpAR10P) dataset to run through our algorithm for feature selection and subsequent clustering. The

Table 3: The outputs related to the ablation study.

| Datasets | GLIOMA | | warpAR10P | |
|---|---|---|---|---|
| Evaluation metrics | ACC± std | NMI± std | ACC± std | NMI± std |
| Main experiment | **54.10 ± 2.71(30)** | **32.09±2.14(40)** | **46.66 ± 3.51(10)** | **48.63 ± 2.83(10)** |
| $\alpha = \beta = \gamma = 0$ | 44.51±2.96(50) | 21.38±4.67(50) | 32.27±2.75(50) | 34.01±3.18(50) |
| $\alpha = \beta = 0$ | 50.21±4.29(20) | 27.09±3.09(70) | 39.65±3.18(10) | 42.59±4.01(30) |
| $\alpha = \gamma = 0$ | 48.30±3.13(10) | 27.52±4.03(60) | 38.15±3.94(30) | 41.55±3.78(80) |
| $\beta = \gamma = 0$ | 46.60±2.16(10) | 25.46±5.29(50) | 40.84±3.06(10) | 45.01±3.56(30) |
| $\alpha = 0$ | 50.00±3.94(40) | 30.37±1.25(10) | 43.34±4.13(20) | 45.68±3.36(10) |
| $\beta = 0$ | 49.80±1.82(30) | 29.89±2.28(30) | 42.19±4.97(20) | 44.04±3.58(20) |
| $\gamma = 0$ | 51.20±2.46(20) | 28.97±4.18(70) | 40.26±3.51(30) | 43.99±2.19(10) |

clustering metrics for the seven experiments are reported in Table 3 and are compared against the best results we obtained with non-zero hyperparameter values of our proposed method. The results indicate that, all parts of the objective function in (9) have a critical role in the feature selection process and can not be removed from this process.

The results of the sensitivity analysis on two datasets, ORL (image) and TOX171 (biological), indicate that while our model's performance depends on the values of the parameters and datasets, it demonstrates relatively consistent ACC and NMI values. Notably, our model operates well with moderate values of the hyperparameters. For detailed information on the sensitivity analysis, refer to A.7 in the Appendix.

## 3.3 Application to the PneumoniaMNIST Dataset

In patients diagnosed with pneumonia, chest X-rays reveal various diagnostic features, particularly in the cardiac section. Notable among these characteristics are an augmented cardiac shadow, alterations in cardiac positioning, the presence of effusion, the pericardial effusion, and the instances of cardiomegaly Reed (2017). In this section, the application of our proposed feature selection method to the PneumoniaMNIST dataset Yang et al. (2021) is assessed with the goal of precisely identifying and analyzing the cardiac silhouette in chest X-ray images. The PneumoniaMNIST dataset, designed for Chest X-Ray radiographs and forming part of the MedMNIST collection Yang et al. (2023), comprises 5,856 Chest X-Ray radiograph images, each measuring

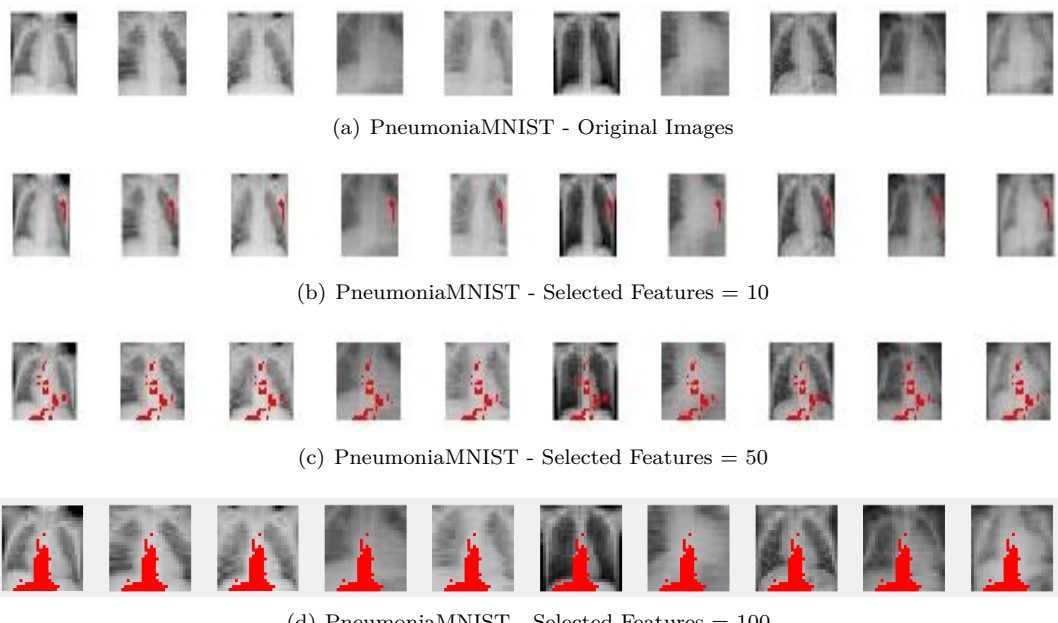

(a) PneumoniaMNIST - Original Images

(b) PneumoniaMNIST - Selected Features = 10

(c) PneumoniaMNIST - Selected Features = 50

(d) PneumoniaMNIST - Selected Features = 100

Figure 3: The visualization of selected features on PneumoniaMNIST images.

28x28 pixels in grayscale. To this end, we test the representativeness of the selected features using GRSSLFS by running it on the PneumoniaMNIST image database. Here, the first ten Chest X-Ray radiograph images are chosen from this dataset. Moreover, we select $\{10, 50, 100\}$ pixels by running GRSSLFS. Figure 3 shows the outputs, where the original and the feature-selected images for each Chest X-Ray radiograph image are plotted in the corresponding row from left to right. In Figure 3, the selected features are depicted by red dots and the un-selected features remain at their original pixel colour. As can be seen, when the number of selected features increases from 10 to 100, subsequently GRSSLFS is able to capture the most *representative* parts of the images, like the central part of the chest, which contains the most discriminative features of the Chest X-Ray radiograph images and can detect the cardiac silhouette. The performance of the obtained feature-selected images has been validated through *assessment by radiologists*, which demonstrates the proficiency of our proposed method in capturing the fundamental features of the PneumoniaMNIST dataset. This capability shows the *interpretability and explainability* power of our model. The reason for this selection may lie in the cooperation of the basis matrix in self-representation and subspace learning terms of our objective function. Finally, Section A.11 reports the results pertaining to the selection of 100 features by other comparative methods. Examining this figure, it is evident that some methods, including our proposed method, SLAP, and VCSDFS, exhibit a tendency to identifythe central area of chest X-ray images. Conversely, some other methods, while successfully identifying portions of the central areas, may overlook additional regions that could provide valuable information. These methods fail to capture such information in radiological analysis.

## 4 Conclusion

In this paper, we addressed the self-representation learning problem by establishing a basis for the feature space, with the primary aim of eliminating unnecessary and redundant features for the representation of the initial data. The motivation behind this work stems from a fundamental concept in linear algebra known as the basis for a given vector space. Essentially, this concept implies that every feature vector can be expressed as a linear combination of a basis set of features, which are linearly independent. Leveraging this idea, we also designed a subspace learning framework to select features with minimal redundancy. Through the introduction of these novel variations in self-representation and subspace learning, we proposed an unsupervised feature selection method called Graph Regularized Self-Representation and Sparse Subspace Learning (GRSSLFS).

Comparative experiments conducted in this study demonstrated that the proposed GRSSLFS method exhibits a high level of effectiveness in feature selection.

The performance of our proposed GRSSLFS method is closely tied to the choice of the feature space basis. As discussed, GRSSLFS employs the variance information of features to establish a basis for the feature space. However, it raises the question: what alternative mathematical or statistical tools could be used instead of variance information for constructing such a basis? This prompts us to recognize the challenge of devising a meaningful basis for the feature space through a potentially more optimized process. As such, the pursuit of constructing an optimal basis for the feature space remains an open problem within the domain of unsupervised feature selection, warranting further research in future studies.

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

## A  Appendix

### A.1  Related Work

Table 1 presents a detailed summary of various unsupervised feature selection methods, systematically classified into the two main categories: Subspace Learning and Self-representation. Both categories are crucial for dimensionality reduction and offer ways to enhance the efficiency and clarity of feature selection models without the need for labeled data. Subspace learning methods aim to identify a lower-dimensional representation of data that maintains its inherent structure, thereby enabling more efficient feature selection. Conversely, self-representation methods strive to represent each data point or feature as a combination of other points or features within the dataset, revealing the underlying relationships and supporting effective feature selection. To further enhance these methodologies, various regularization and constraint techniques can be applied. Graph regularization utilizes the geometric structure of the data, ensuring that the selected features preserve the inherent relationships among data points. Sparse regularization encourages the selection of a minimal subset of features, thereby improving the efficiency and interpretability of model. Meanwhile, the orthogonality constraint ensures that the selected features are uncorrelated, promoting a more robust and diverse feature set. By incorporating these regularization and constraint techniques with subspace learning-based and self-representation-based methods, researchers can attain more robust, efficient, and interpretable feature selection outcomes.

Table 1 lists some of these methods and their potential combinations, providing a valuable resource for advancing the field of unsupervised feature selection. To begin with, Wang et al. (2015a) introduced the MFFS method, which is a notable example of unsupervised feature selection developed using subspace learning and matrix factorization techniques. Subsequently, Wang et al. (2015b) proposed the MPMR method, a subspace learning-based approach that utilizes maximum projection and minimum redundancy. This method aims to capture the most significant variance while ensuring that the selected features provide complementary information, thus enhancing clustering and classification performance. Building on subspace learning and matrix factorization, Qi et al. (2018) introduced the RMFFS method. This technique employs regularized matrix factorization for feature selection, effectively decomposing the data into lower-dimensional matrices and facilitating the identification of informative features while imposing an inner product regularization constraint to ensure orthogonality. Tang et al. (2018) introduced a robust unsupervised feature selection approach, called DSRMR, which leverages dual self-representation and manifold regularization. By capturing both local and global data structures, this method ensures that the selected features preserve the intrinsic geometry of the data. Inspired by MFFS, Zheng et al. (2019) suggested the NSSLFS method, which utilizes the nonnegative subspace learning combined with the $L_1$ sparsity to select highly discriminative features. Parsa et al. (2020) introduced SCFS, a non-linear self-representation-based method that incorporates the adaptive similarity learning and subspace clustering techniques. SCFS dynamically adjusts similarity measures to enhance the capture of feature relationships. Recent advancements have continued to advance the unsupervised feature selection methods. For example, Liu et al. (2020) introduced the RNE method, a robust neighborhood subspace learning technique designed to preserve local neighborhood structures and enhance feature selection quality, particularly in noisy environments. Shang et al. (2020) and Sheng et al. (2021) introduced the SLSDR and DGSLFS methods, respectively. These feature selection techniques utilize dual-graph regularization within the subspace learning framework to capture data structures effectively.

Further extending these advancements, Yuan et al. (2022) proposed the CNAFS method, which utilizes convex non-negative matrix factorization alongside an adaptive graph. CNAFS ensures that feature selection maintains data manifold structures through a resilient and adaptive subspace learning strategy. Li et al. (2022) developed SPLR, a subspace learning method that is based on the principles of self-paced learning. SPLR seeks to preserve the local manifold structure and utilizes the $L_{2,1/2}$-norm to maintain discriminative features while reducing the impact of noise in the data. Chen et al. (2022) introduced RDMRS2FS, a feature selection technique rooted in self-representation. RDMRS2FS focuses on semi-supervised feature selection

with the goal of reducing redundancy among features. This method utilizes dual-graph regularization to precisely capture data characteristics and uncover relationships between features.

## A.2 Datasets

Table 4 demonstrates the eight benchmark datasets, which are used to perform our experiments. We selected a mix of five different gene expression datasets and three facial image datasets to diversify our data samples and hence to better evaluate the performance of our proposed feature selection method. It should be noted that all of these datasets are accessible through the scikit-feature feature selection repository[*] Li et al. (2018) and the UCI Machine Learning Repository[†] Dua & Graff (2017).

Table 4: The information of eight datasets used in this study.

| ID | Dataset | # of Samples | # of Features | # of Classes | Type |
|----|---------|--------------|---------------|--------------|------|
| 1 | **CNS** | 60 | 7129 | 2 | Biological |
| 2 | **GLIOMA** | 50 | 4434 | 4 | Biological |
| 3 | **TOX_171** | 171 | 5748 | 4 | Biological |
| 4 | **SRBCT** | 83 | 2308 | 4 | Biological |
| 5 | **SMK** | 187 | 19993 | 2 | Biological |
| 6 | **ATT** | 400 | 10304 | 40 | Image |
| 7 | **ORL** | 400 | 1024 | 40 | Image |
| 8 | **warpAR10P** | 130 | 2400 | 10 | Image |

## A.3 Comparison Methods

In this study, the following feature selection methods are selected with their track records of performance in the literature to be compared with GRSSLFS. These methods are briefly explained below.

1. **Baseline**: All features of data are considered.

2. **The least absolute shrinkage and selection operator (Lasso)** Tibshirani (1996): A supervised feature selection method that efficiently selects features by assuming a linear relationship between input features and output values.

3. **Laplacian Score (LS)** He et al. (2005): The features selected by this method can best preserve the local manifold structure of the original data.

4. **Convex non-negative matrix Factorization with an Adaptive graph constraint Feature Selection (CNAFS)** Yuan et al. (2022): Through convex matrix factorization with adaptive graph constraint, it selects the best feature subset by considering the correlation between the data while preserving the local manifold structure.

5. **Orthogonal basis clustering and Local Structure Preserving (OCLSP)** Lin et al. (2022): An orthogonal basis clustering and an adaptive graph regularization are used to gain cluster separation while keeping the local information of data.

6. **Supervised Feature Selection with Local Adaptive Projection (SLAP)** Chen et al. (2018b): is a supervised feature selection method that simultaneously learns an adaptive similarity matrix and a projection matrix within an iterative framework.

7. **Robust Neighborhood Embedding (RNE)** Liu et al. (2020): In this method, features are selected using the neighborhood embedding and $\ell_1$ norm as the loss function.

---

[*] `https://jundongl.github.io/scikit-feature/datasets.html`
[†] `https://archive.ics.uci.edu/ml/index.php`

8. **Variance-Covariance Subspace Distance based Feature Selection (VCSDFS)** Karami et al. (2023): Variance-Covariance subspace distance is used to select the subset which contains the most representative features.

9. **Concrete Autoencoders (CAE)** Balın et al. (2019): CAE has an encoder-decoder architecture with a concrete selector layer as the encoder and a standard neural network as the decoder. It aims at selecting the most informative subset of global features, which are used to simultaneously train a neural network to reconstruct the input data.

10. **Coordinate Descent based Least Square Regression (CD-LSR)** Xu et al. (2023): This method leverages a coordinate descent-based optimization framework to solve the general $l_{2,0}$-norm constrained feature selection problem.

## A.4 Evaluation Metrics

In the experiments, the clustering performance of feature selection algorithms is evaluated using Accuracy (ACC) and Normalized Mutual Information (NMI) metrics Solorio-Fernández et al. (2020). The higher ACC and NMI values are, the better the clustering performance is, which in turn results from better performance of the feature selection algorithm to select informative features.

The ACC is defined as:

$$\text{ACC} = \frac{\sum_{i=1}^{n} \delta(q_i, \text{map}(p_i))}{n},$$

where $p_i$ and $q_i$ are, respectively, the clustering and the ground-truth labels of the sample $x_i$, and $\delta$ is the Kronecker delta function. In addition, map(.) denotes the best permutation mapping function that uses the Kuhn–Munkres algorithm to map each clustering label to the corresponding ground-truth label.

The NMI is defined as

$$\text{NMI} = \frac{MI(P, Q)}{\sqrt{H(P)H(Q)}},$$

where $P$ and $Q$ are the set of centers of the predicted and the ground-truth clusters, and $MI(.)$ is the mutual information. Furthermore, $H(P)$ and $H(Q)$ denote the entropy of $P$ and $Q$, respectively.

## A.5 Experimental Settings

Our experiments are performed for various numbers of selected features in the range of $\{10, 20, \ldots, 100\}$, for all datasets. Furthermore, hyperparameter tuning is done to find the most optimum configuration of various feature selection models in our study. We fix $k = 5$ for the $k$-nearest neighbor algorithm and for all the datasets in LS, OCLSP and RNE methods. For CNAFS, the hyperparameters $\alpha, \beta, \gamma, \lambda$ and $\epsilon$ are tuned by searching from $\{10^{-3}, 10^{-2}, \ldots, 10^3\}$ as mentioned in author's literature in lYuan et al. (2022). To implement OCLSP, Lin et al. (2022), the parameter $\alpha$ is fixed $10^4$ and $\eta, \gamma, \beta$ are tuned by selecting from $\{10^{-3}, 10^{-2}, \ldots, 10^3\}$. For SLAP, we set the projected dimension $m$ equal to the number of classes, and the regularization parameter $\gamma$ is selected from the set $\{10^{-3}, 10^{-2}, \ldots, 10^3\}$. In VCSDFS, the values of the parameter $\rho$ is selected from $\{10^{-6}, 10^{-5}, \ldots, 10^6\}$, Karami et al. (2023). We adopted the values for the parameters of RNE from Liu et al. (2020), and let the penalty coefficient $\alpha$ be $10^3$. For the CAE method, we used the values in the original paper Balın et al. (2019). The performance of CD-LSR was assessed using the provided code based on the author's implementation as described in the original paper of the method Xu et al. (2023). We tuned the parameters $\alpha, \beta$ and $\gamma$ of GRSSLFS method on $\{10^{-5}, \ldots, 10^5\}$. Finally, due to the random initialization in the $k$-means clustering algorithm, we repeated the clustering runs 20 times with different random starting points over which we reported the average and standard deviations of the ACC and NMI metrics. Finally, it should be noted that our tests were conducted utilizing Matlab 2018a on a personal computer equipped with 16GB of RAM and an Intel Core i5-4690 processor.

## A.6 Experimental Results

For more illustration of the reported clustering performance of all comparison methods, in Figures 4 and 5, we depict the values of ACC and NMI metrics for different number of selected features.

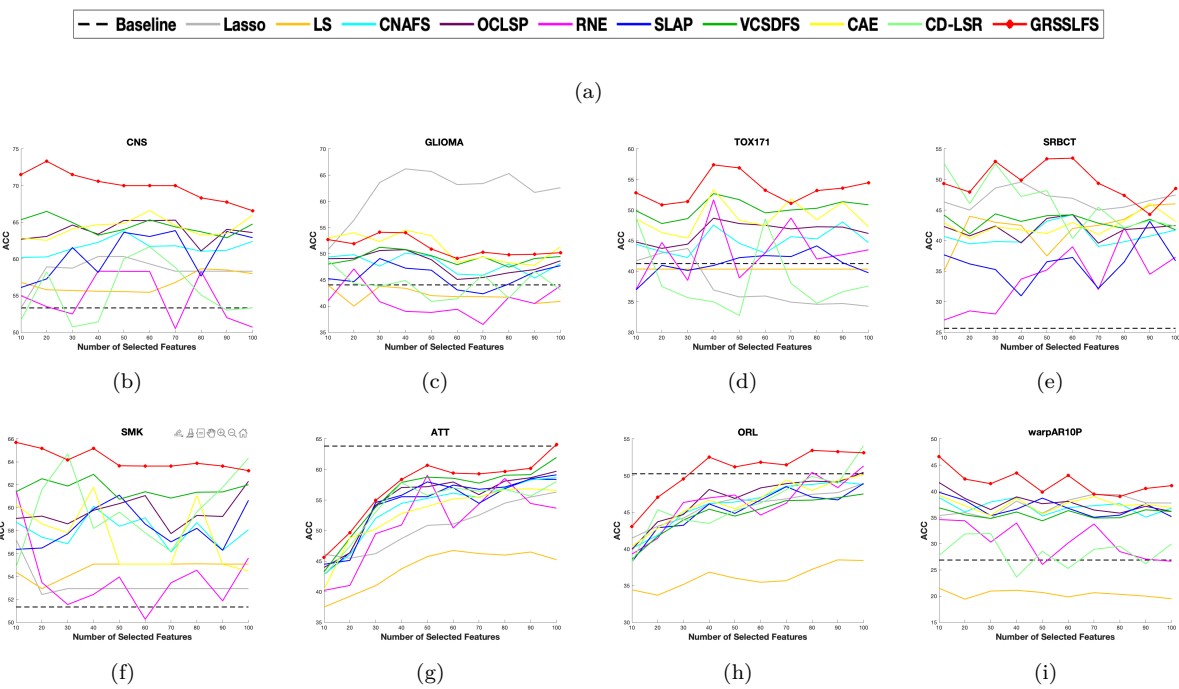

Figure 4: Clustering ACC of 11 algorithms across eight datasets as a function of the number of selected features.

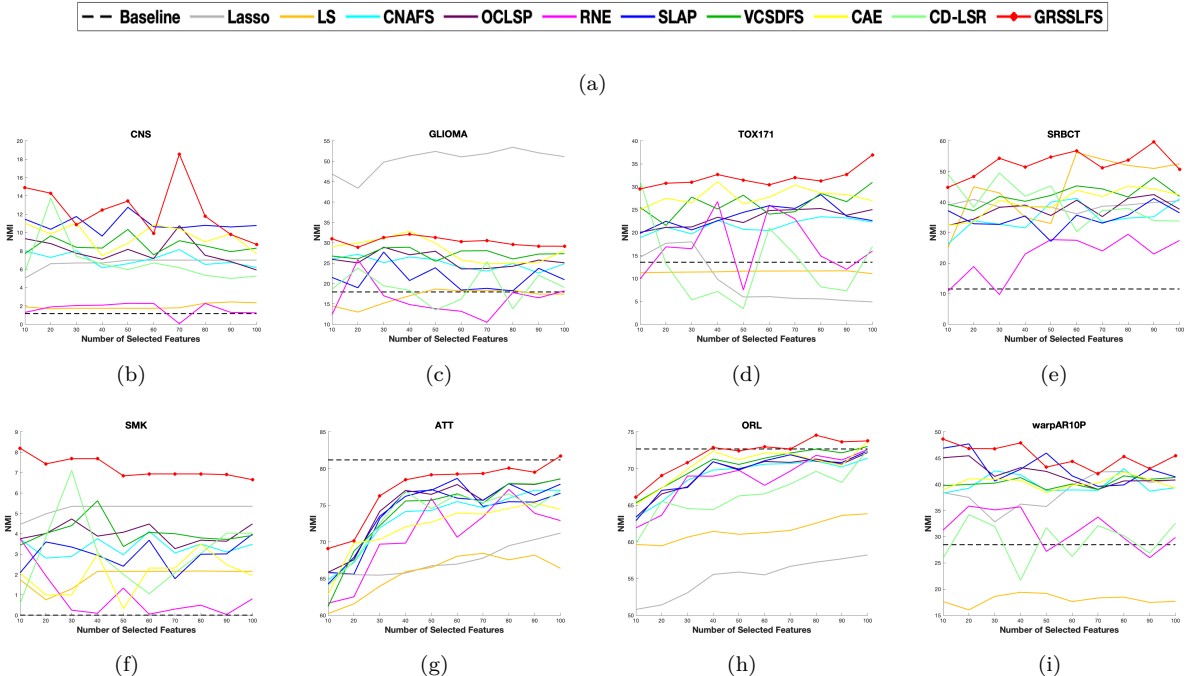

Figure 5: Clustering NMI of 11 algorithms across eight datasets as a function of the number of selected features.

## A.7 Sensitivity Analysis

We conducted a sensitivity experiment on one image dataset, ORL, and one biological dataset, TOX171. We fixed the number of selected features at $k = 100$. Since we have three hyperparameters for evaluating sensitivity, we fixed one parameter and calculated the values of ACC and NMI with respect to the other two varying hyperparameters. We set the value of the fixed parameter to $10^{-1}$ for this experiment. The results are illustrated in Figure 6. It is evident that our model has relatively steady performance in terms of the hyperparameters, and the results don't show high fluctuations. As an observation, we can say that GRSSLFS produces better outputs for moderate values of the hyperparameters in the range of $10^{-2}$ to 10.

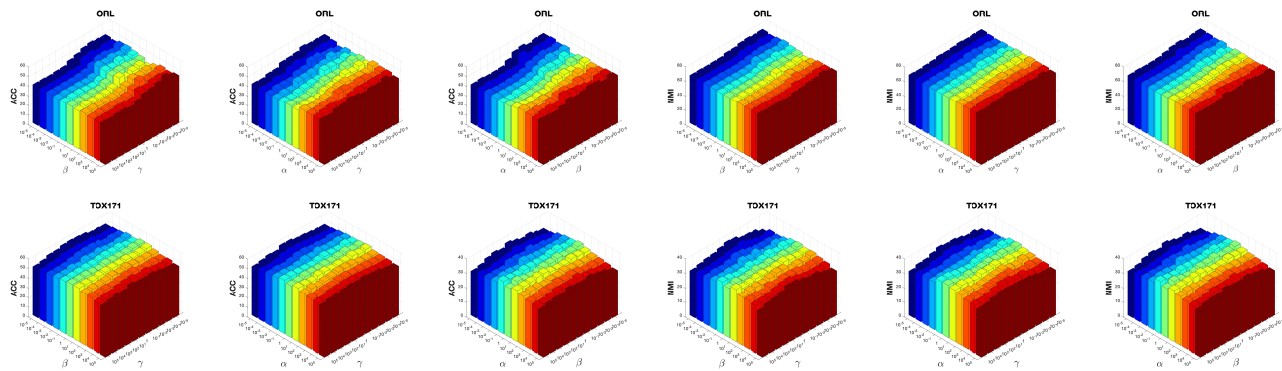

Figure 6: Sensitivity experiment: impact of the hyperparameters $\alpha$ and $\beta$, and $\gamma$ on the values of ACC and NMI for the datasets ORL and TOX171.

## A.8 Running Time Comparison

Figure 7 also demonstrates a comparison between the proposed method and all other unsupervised feature selection algorithms in terms of the running times (in seconds) for selecting 100 features from the TOX171 dataset. It is evident that CAE is the slowest method, followed by RNE and GRSSLFS methods, as the second and third slowest algorithms. The high running time of CAE is ascribed to the training neural networks in this deep, encoder-decoder feature selector Balın et al. (2019). The relative sluggishness of the RNE method can be attributed to its non-smooth nature, although it is convex Liu et al. (2020). To solve this issue, RNE employs the alternation direction method of multipliers (ADMM) to minimize its loss function. The proposed method is the next slow method due to the fact that the GRSSLFS method uses both subspace

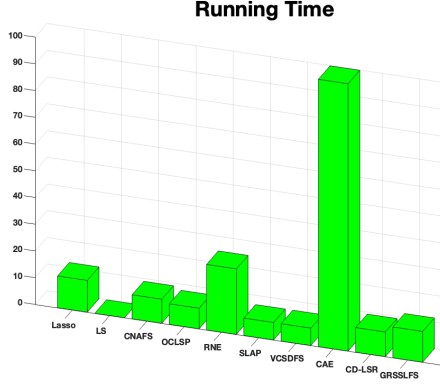

Figure 7: Running time (in seconds) of all algorithms to select 100 features for the TOX171 dataset.

learning and self-representation, wherein multiple matrix multiplications are carried out. Consequently, as we demonstrated in Section 2.7, the time complexity of the GRSSLFS method is quadratic in terms of both the number of original features and the number of input data points. Furthermore, as we discussed in Section 2.5, an alternative approach is used to solve the loss function minimization problem of the proposed method.

## A.9  Convergence Test

We set out to empirically evaluate Theorem 2.2, wherein we presented a theoretical proof for the monotonic decreasing behavior of the objective function of the GRSSLFS method. Figure 8 depicts the evolution of the objective function over the number of iterations for all datasets. One can clearly see that the values of the objective function across all datasets plummet quickly and the objective function converges quickly. This observation demonstrates the effectiveness of the alternative iterative algorithm that is proposed to solve the objective function of the GRSSLFS method.

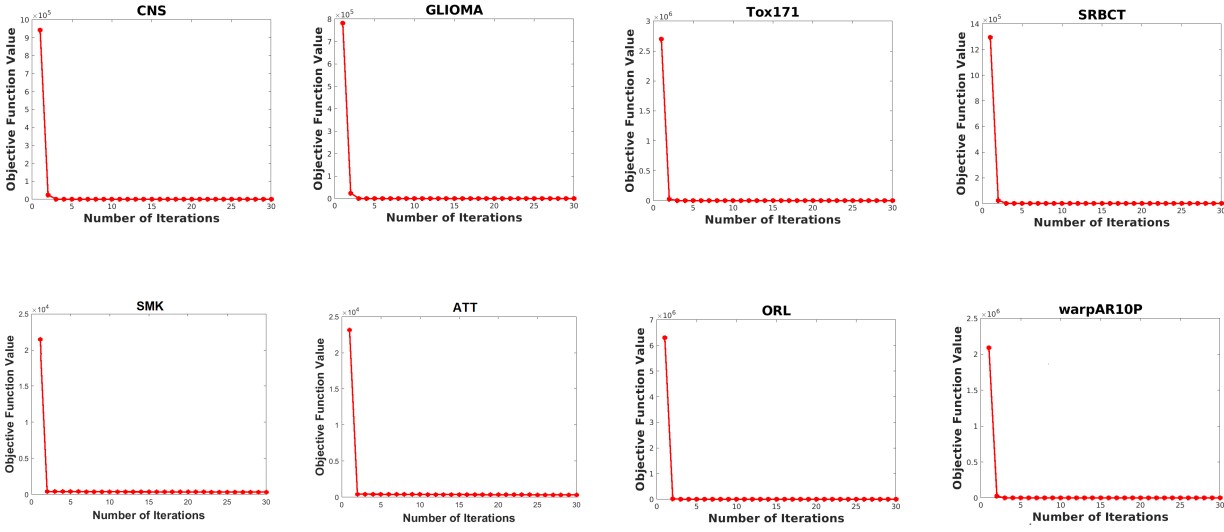

Figure 8: The convergence curves of the objective function of GRSSLFS on the eight datasets.

## A.10  Non-parametric Statistical Test

We demonstrated in previous sub-sections that the proposed method, GRSSLFS, has superior performance over the state-of-the-art unsupervised feature selection methods. Here, we employ the Friedman test as a non-parametric statistical test to evaluate the statistical significance of GRSSLFS's performance compared to other methods. The null hypothesis is then expressed as follows: there is no significant difference between any of the methods, and the selected features by these methods lead to the same clustering performance. The alternative hypothesis is that there are at least two methods whose selected features result in significant differences in the clustering performance.

We use the data presented in Table 2 to run the test. In this setting, the datasets, the feature selection methods, and the ACC or NMI values are considered as, respectively, the subjects, the treatments, and the measurements in the Friedman test terminology. The feature selection methods are then ranked to calculate the Friedman statistics. The average rankings over all datasets for each feature selection algorithm and two different measurements (i.e., ACC and NMI) are presented in Figure 9. The lower ranking means the corresponding measurement is higher.

We obtained 38.022727 and 36.090909 for the Friedman statistics based on ACC and NMI measurements, respectively. Accordingly, the $p$-values corresponding to these Friedman statistics are 0.000038 and 0.000081,

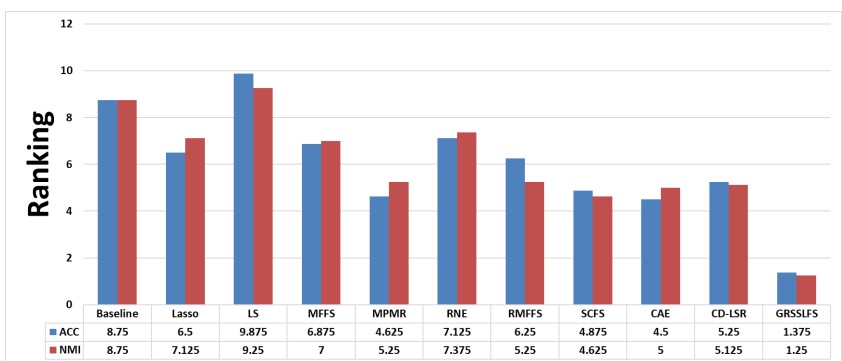

Figure 9: Average ranks obtained by the Friedman test for each method with respect to the ACC and NMI evaluation metrics on the datasets. Note that the lower rank of the evaluation metrics, the better the performance of the methods.

respectively. With the significance level of 5%, these two $p$-values lead us to reject the null hypothesis in the favor of the alternative hypothesis. With this description, it can be seen that there are at least two methods, VCSDFS, CAE and GRSSLFS, that select features with statistically significant differences in the clustering performance.

Following the Friedman test, we also carry out a post-hoc statistical test using the Holm's method (aka., Holm–Bonferroni method) Bolón-Canedo et al. (2014) to identify which pairs of methods result in a significant difference in the clustering performance. For this purpose, we select the proposed method, GRSSLFS, as the control method and compare it with all other 10 methods including the Baseline which are listed in the columns of Table 2. It means that we end up having 10 null hypotheses for each pair of GRSSLFS-X where "X" refers to those 10 other methods. Each null hypothesis states that there is no significant difference between GRSSLFS and the method "X" in terms of their feature selection effectiveness with performant downstream clustering. The significance level $\alpha$ is set to 0.05 for each null individual null hypothesis. The $p$-values of the nine null hypotheses are presented in Table 5 based on ACC and NMI measurements, respectively. The corresponding Holm's $p$-values are also included in these tables.

| ACC | | |
|---|---|---|
| Method | $p$-value | Holm's $p$-value |
| LS | 0 | 0.005 |
| Baseline | 0.000009 | 0.005556 |
| RNE | 0.000526 | 0.00625 |
| CNAFS | 0.000911 | 0.007143 |
| Lasso | 0.001998 | 0.008333 |
| SLAP | 0.003285 | 0.01 |
| CD-LSR | 0.019454 | 0.0125 |
| VCSDFS | 0.034808 | 0.016667 |
| OCLSP | 0.050016 | 0.025 |
| CAE | 0.059505 | 0.05 |

| NMI | | |
|---|---|---|
| Method | $p$-value | Holm's $p$-value |
| LS | 0.000001 | 0.005 |
| Baseline | 0.000006 | 0.005556 |
| RNE | 0.000221 | 0.00625 |
| Lasso | 0.000396 | 0.007143 |
| CNAFS | 0.000526 | 0.008333 |
| OCLSP | 0.015861 | 0.01 |
| SLAP | 0.015861 | 0.0125 |
| CD-LSR | 0.019454 | 0.016667 |
| CAE | 0.023739 | 0.025 |
| VCSDFS | 0.041831 | 0.05 |

Table 5: Post-hoc test using Holm's method to compare the effect of GRSSLFS and the 10 other feature selection methods on the clustering ACC and NMI metrics. The significance level of $\alpha = 0.05$ is set for each individual pairwise null hypothesis. Holm's procedure rejects a null hypothesis when the p-value of the individual null hypothesis is less than or equal to Holm's $p$-value.

Holm's method rejects those hypotheses that their initial $p$-values are less than or equal to the calculated $p$-values by Holm's method. Thus, we infer from the results in Table 5 that we have enough evidence to reject the null hypothesis for the pairwise comparison between GRSSLFS and all methods except for SLAP, OCLSP, CAE, VCSDFS, and CD-LSR. To be more precise, our proposed GRSSLFS method exhibits significant differences from all other methods, except for the aforementioned methods, in terms of the ACC and NMI measures. Nevertheless, as we observed in the results of Figures 4 and 5, GRSSLFS displayed a more

consistent trend over a wide range of a number of selected features compared to all comparison methods. In addition, although there is no significant difference between our proposed GRSSLFS method (an unsupervised approach) and the two supervised approaches, SLAP and CD-LSR, the numerical results demonstrate that the proposed unsupervised feature selection method in this article is fully capable of competing with these supervised methods.

### A.11 Application to the PneumoniaMNIST dataset

This section presents the results of the selection of 100 features from the PneumoniaMNIST dataset, conducted by our proposed GRSSLFS method and other comparative methods (see Figure 10).

### A.12 Code Availability

The source code is available at `https://github.com/FaridSaberi/GRSSLFS`.

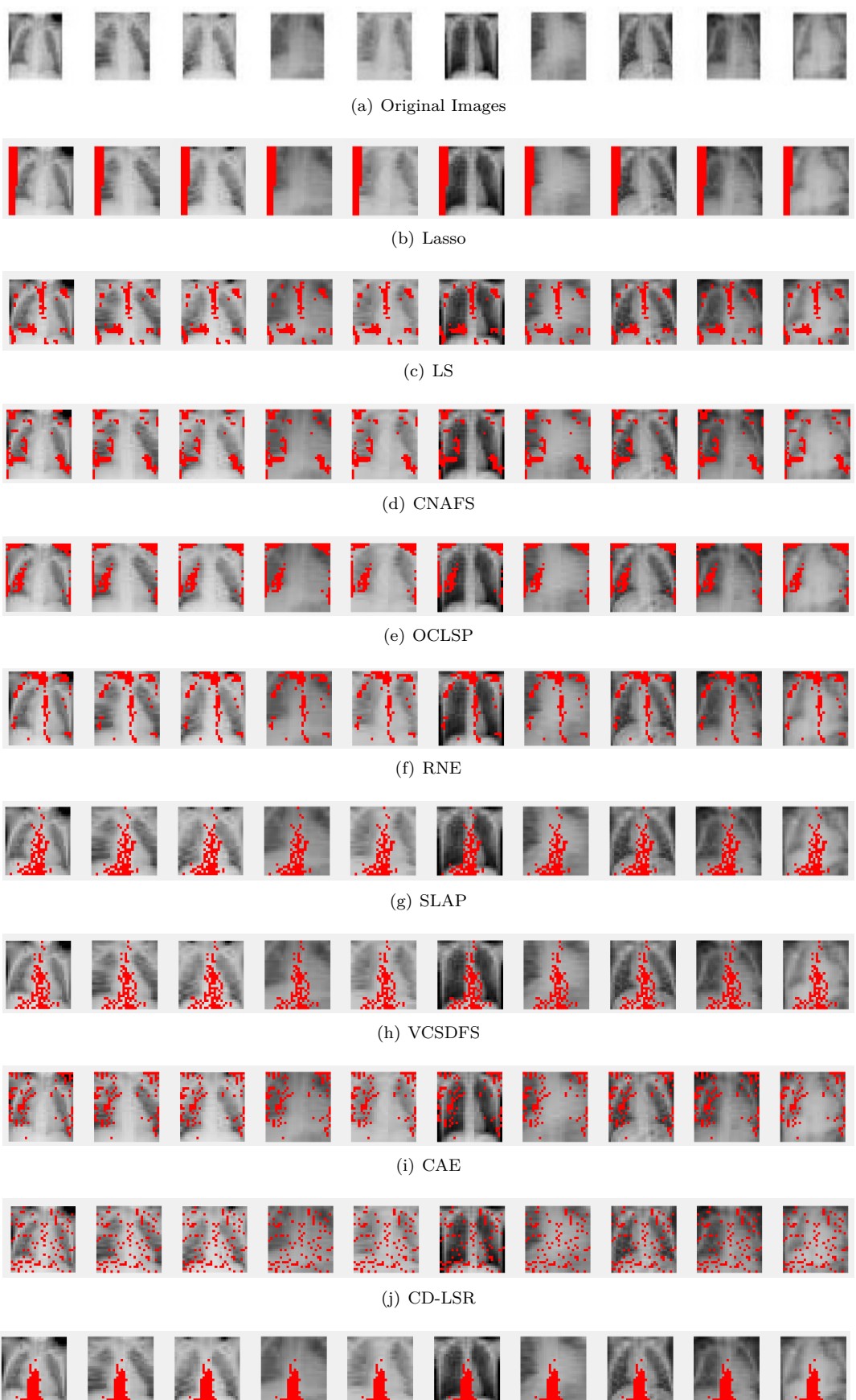

(a) Original Images

(b) Lasso

(c) LS

(d) CNAFS

(e) OCLSP

(f) RNE

(g) SLAP

(h) VCSDFS

(i) CAE

(j) CD-LSR

(k) Our proposed GRSSLFS method

Figure 10: The visualization of 100 selected features obtained by different feature selection methods on PneumoniaMNIST images.