# OpenReview forum: "A Self-Representation Learning Method for Unsupervised Feature Selection using Feature Space Basis"
_TMLR — Accepted by TMLR_

### Review · Reviewer_3sXe · 2024-05-13

**Summary Of Contributions:**

This work proposes a linear feature selection method based on the feature space basis

**Audience:**

Yes

**Broader Impact Concerns:**

Fine

**Claims And Evidence:**

Yes

**Requested Changes:**

See weakness 2.

**Strengths And Weaknesses:**

The main strength is that that the method is evaluated against many other baseline methods across many datasets. To me, this part of the paper seems convincing.

There are two weaknesses:
1. The method is only learning in the linear features, but I guess this is not a fixable point
2. The motivation feels unclear. Why do unsupervised feature selection when there is already quite good supervised feature selection method? I think this point is related to the fact that the paper do not compare with any supervised feature selection method (such as straightforward lasso). In the tables, I think the authors should also show the performance of supervised feature selection methods and convince the readers that using unsupervised learning methods give competitive results (if not better)

---

> ### Author Response · Authors · 2024-06-13
>
> >#### Comment 1
> > The method is only learning in the linear features, but I guess this is not a fixable point.
>
> Thank you for your insightful comment regarding the linearity of the feature learning in our proposed model. Our objective function comprises three distinct terms, each designed to address specific aspects of the data and the feature selection process. Here is how each term contributes to the proposed model:
>
> 1. The term **self-representation problem based on the basis**, which is defined as:
> $$\min_{\mathbf{G}\in\mathbb{R}^{m\times n}}\Vert\mathbf{X}-\mathbf{BG}\Vert_F^2.$$
> This term is a linear term that aims to reconstruct the data $\mathbf{X}$ by linearly combining the basis vectors. This ensures that the selected features can linearly approximate the original data, which can preserve the essential information. In this regard, this choice was motivated by two factors:
>
> a) Simplicity and Interpretability: Linear methods offer clear interpretability, which is crucial for understanding the underlying relationships in the data and for practical applications where interpretability is of utmost importance.
>
> b) Computational Efficiency: Linear models are generally less computationally intensive compared to their non-linear counterparts, making them suitable for large-scale datasets.
>
> However, as you mentioned, linear methods can be limited in their ability to capture complex and non-linear relationships. To overcome this challenge, we have incorporated the following terms into our proposed objective function.
>
> 2. The term **subspace learning problem based on the basis**, which is defined as:
> $$\min_{\mathbf{G}, \mathbf{U}, \mathbf{V}}\Arrowvert \mathbf{B}-\mathbf{BGUV}\Arrowvert_F^2.$$
>
> This subspace learning component is bi-linear, involving both $\mathbf{B}$ and the product
> $\mathbf{BGUV}$. This term ensures that the basis
> $\mathbf{B}$ can be represented within a learned subspace, which is governed by the matrices
> $\mathbf{G}$, $\mathbf{U}$, and $\mathbf{V}$. This interaction introduces a bi-linearity in the feature learning process by enabling the basis to adapt within a subspace that captures more complex relationships in the data. It should be noted that, despite having linear interaction terms in a bilinear model, bilinear models can capture non-linear relationships due to the product of features.
>
> 3. The term **feature graph regularization**, which is defined as:
> $$\min_{\mathbf{G}}\mathrm{Tr}\left(\mathbf{B}\mathbf{G}\mathbf{L}\mathbf{G}^T\mathbf{B}^T\right).$$
> This feature graph regularization term is inherently non-linear. It leverages the Laplacian matrix $\mathbf{L}$, which encodes the geometric properties of the feature space, to ensure that the learned features preserve these properties. To be more specific, the expression
> $\mathrm{Tr}\left(\mathbf{B}\mathbf{G}\mathbf{L}\mathbf{G}^T\mathbf{B}^T\right)$ is a quadratic form and is not a linear function. It is known that quadratic forms represent a higher-order relationship than linear functions, specifically involving squared terms or products of terms.
> As a result, the use of the term $\mathrm{Tr}\left(\mathbf{B}\mathbf{G}\mathbf{L}\mathbf{G}^T\mathbf{B}^T\right)$
> introduces non-linearity, capturing higher-order dependencies and interactions among features.
>
> In summary, while the  *self-representation problem based on the basis* is a linear model in our problem, the inclusion of the bilinear term *subspace learning problem based on the basis* and the nonlinear term *feature graph regularization* can significantly enhance the model's capability to capture complex patterns and relationships within the data. These terms collectively ensure that the learning process is not limited to linear relationships but can also model higher-order interactions and preserve the geometric structure of the feature space.
>
> We hope this explanation clarifies the design and intention behind our model, demonstrating that it incorporates both linear and nonlinear aspects to provide a comprehensive and robust feature learning framework.

---

> ### Author Response · Authors · 2024-06-14
>
> >#### Comment 2:
> > The motivation feels unclear. Why do unsupervised feature selection when there is already quite good supervised feature selection method? I think this point is related to the fact that the paper do not compare with any supervised feature selection method (such as straightforward lasso). In the tables, I think the authors should also show the performance of supervised feature selection methods and convince the readers that using unsupervised learning methods give competitive results (if not better).
>
> We sincerely appreciate the reviewer's insightful comments. In response, we would like to clarify that our aim is to introduce a new unsupervised feature selection (FS) model. It is generally expected that supervised FS models perform better than unsupervised FS models, as they utilize label information from the data. However, since the labeling process is costly and often time-consuming,  as the reviewer mentioned, it is valuable to design unsupervised FS models that can compete with supervised FS methods.
>
> In the initial version of the paper, we included a supervised FS method in the numerical experiments: the CD-LSR method. To address the reviewer's suggestion, we have now added two other supervised FS methods, LASSO and SLAP, to our comparison methods. Specifically, we have included:
>
> The Least Absolute Shrinkage and Selection Operator (LASSO),
>
> Supervised Feature Selection with Local Adaptive Projection (SLAP),
>
> Coordinate Descent-based Least Squares Regression (CD-LSR).
>
> For more information about these models, please refer to Subsection A.3 Comparison Methods. It should be noted that comparing our proposed GRSSLFS method (an unsupervised approach) with other supervised methods, such as Lasso, SLAP, and CD-LSR, reveals that our unsupervised feature selection technique is fully capable of competing with these supervised alternatives, as reported in Table 2. However, on the GLIOMA dataset, the performance of LASSO is better than that of all unsupervised models, including ours. For more information, we kindly ask the reviewer to refer to Section 3.1 Results and Analysis on Pages 10 and 11 of the revised manuscript.

---

### Review · Reviewer_X5pW · 2024-05-18

**Summary Of Contributions:**

To solve the problem of redundant features in the representation space, the authors propose a new unsupervised feature selection method called GRSSLFS, which integrates a basis of linearly independent features with the highest variance scores into a unified framework of subspace learning and self-representation. Both theoretical analysis and experimental results demonstrate the effectiveness of the method.

**Audience:**

Yes

**Broader Impact Concerns:**

The work does not raise any concerns related to broader societal or ethical impacts.

**Claims And Evidence:**

Yes

**Requested Changes:**

- The authors could provide a detailed explanation of the term "Graph Regularized." Specifically, it is important to describe how graph regularization is integrated into the method, what role it plays in enhancing the feature selection process, and how it helps to preserve the geometric structure of the data, which would strengthen the work.
- The article can include a section dedicated to related work. In this section, the authors can discuss the various unsupervised feature selection methods and their key characteristics as outlined in Table 1, which would provide a comprehensive overview of existing approaches. This addition would not only strengthen the article but also help readers better understand the significance and novelty of the work.
- The article can include a detailed sensitivity analysis of the hyper-parameters $\alpha$, $\beta$, and $\gamma$. This analysis would examine how variations in these parameters affect the performance of the proposed method across different datasets. By providing this information, the authors can demonstrate the robustness and stability of their method, as well as offer practical guidance on how to tune these hyper-parameters for optimal results, which would strengthen the work.

**Strengths And Weaknesses:**

Strengths:

- The authors propose a new feature selection method called GRSSLFS that merges the subspace learning and self-representation problems, leveraging the basis of the feature space, to simultaneously eliminate redundant data features and feature selection.
- The proposed method is compared with a variety of comparison baselines on multiple datasets. The results demonstrate the effectiveness of the method. The ablation study also shows the necessity of each component.
- Theoretical analysis of the convergence of the method increases the solidness of the article.

Weaknesses:

- The proposed method is called "Graph Regularized Self-Representation and Sparse Subspace Learning" (GRSSLFS). However, the authors do not provide a clear explanation of the term "Graph Regularized," which may lead to confusion among readers who are not familiar with this area.
- The related unsupervised feature selection methods and their key characteristics, as presented in Table 1, can be thoroughly explained. This will provide readers with a comprehensive review of the current state of research in this area.
- The parameters $\alpha$, $\beta$, and $\gamma$ need to be tuned in a large search space $\{10^{-5},...,10^5\}$. This extensive range indicates that the method requires careful adjustment of these hyper-parameters to achieve optimal performance. Consequently, there is a concern regarding the method's sensitivity to these hyper-parameters when applied to various datasets.

---

> ### Author Response · Authors · 2024-06-13
> **Official Comment by Authors**
>
> We are thankful to the reviewer for the careful reading and the valuable feedbacks.
>
> >#### Comment 1:
> > The authors could provide a detailed explanation of the term "Graph Regularized." Specifically, it is important to describe how graph regularization is integrated into the method, what role it plays in enhancing the feature selection process, and how it helps to preserve the geometric structure of the data, which would strengthen the work.
>
> We appreciate the reviewer's insightful comment regarding the incorporation of graph regularization into our feature selection method. We acknowledge that our manuscript currently lacks a clear explanation of how graph regularization is integrated and its significance in enhancing the feature selection process.
>
> To address this, we have revised Subsection 2.4 to provide a detailed description of the term "Graph Regularization" and explain how it is integrated into our objective function. For more information, we kindly ask the reviewer to refer to Subsection 2.4 and the part titled "The Role of $\mathbf{G}$" on Page 6 of the revised manuscript.
>
> In addition to this, we have revised Subsection 2.4 to highlight the role of the feature weight matrix $\mathbf{U}$ and the representation matrix $\mathbf{V}$, which play a major role in selecting the underlying features in our proposed method. For more information, we kindly ask the reviewer to refer to Subsection 2.4 and the parts titled "The Role of $\mathbf{U}$" and "The Role of $\mathbf{V}$" on Pages 5 and 6 of the revised manuscript.
>
>
> >#### Comment 2:
> > The article can include a section dedicated to related work. In this section, the authors can discuss the various unsupervised feature selection methods and their key characteristics as outlined in Table 1, which would provide a comprehensive overview of existing approaches. This addition would not only strengthen the article but also help readers better understand the significance and novelty of the work.
>
> We appreciate the reviewer's insightful comment regarding the inclusion a section to discuss the various unsupervised feature selection methods and their key characteristics as outlined in Table 1. In response to your comment, we have added a new section titled "Related Work" to our paper. This section provides a detailed descriptions of the methods as outlined in Table 1. For more information, we kindly ask the reviewer to refer to Appendix A1 Page 16 of the revised manuscript.
>
> >#### Comment 3:
> > The article can include a detailed sensitivity analysis of the hyper-parameters $\alpha$, $\beta$, and $\gamma$. This analysis would examine how variations in these parameters affect the performance of the proposed method across different datasets. By providing this information, the authors can demonstrate the robustness and stability of their method, as well as offer practical guidance on how to tune these hyper-parameters for optimal results, which would strengthen the work.
>
> We would like to thank the reviewer for bringing this issue to our attention. In response, we have added a subsection titled "Sensitivity Analysis" in Appendix Subsection A7 Page 18. Additionally, we have changed the title of the "Ablation Study" subsection in the main part of the manuscript to "The Role of Hyperparameters." This subsection now includes both our ablation study and explanations about the sensitivity experiment results at the last paragraph which refer to Subsection A7, Page 18. There, we explain that our method is relatively stable with respect to hyperparameters, with the best results obtained for moderate values between 0.01 and 10.

---

### Review · Reviewer_zrhA · 2024-05-31

**Summary Of Contributions:**

I apologize that I didn't review the paper in time and won't be able to do so in the next few days.

**Audience:**

Yes

**Broader Impact Concerns:**

N.A.

**Claims And Evidence:**

Yes

**Requested Changes:**

N.A.

**Strengths And Weaknesses:**

N.A.

---

> ### Author Response · Authors · 2024-06-16
>
> Thank you for letting us know. We understand that schedules can be demanding and appreciate your consideration.

---

### Decision · Action_Editor_Q1TZ · 2024-07-08

**Recommendation:** Accept with minor revision

**Comment:**

This paper proposes a new method for unsupervised feature selection, which has been verified as effective across multiple datasets. Two reviewers have provided positive comments, highlighting the effectiveness of the method. However, one reviewer has raised concerns about the presentation, such as the use of the term "Graph Regularized," the discussion of related work, and the method's sensitivity. The authors are required to revise the paper carefully according to the reviewer's suggestions. Overall, this paper fits the TMLR acceptance criteria, and I recommend accepting it with minor revisions.

**Audience:**

Yes

**Claims And Evidence:**

Yes

---

> ### Author Response · Authors · 2024-07-14
>
> Dear Action Editor,
>
> Thank you for your positive feedback and for recommending our paper for acceptance with minor revisions. We appreciate the thorough evaluation by all reviewers and their constructive comments, which have helped us identify areas for improvement.
>
> In response to the reviewer's concerns, we have already made the following revisions in the latest version of our manuscript:
>
> **(1) Clarification of ``Graph Regularized" Term:** We have revised the section where the term "Graph Regularized" is used to ensure clarity and consistency. We provided a detailed explanation of the term and its significance in the context of our method.
>
> In response to this comment, we have revised Subsection 2.4 to provide a detailed description of the term "Graph Regularization" and explain how it is integrated into our objective function. For more information, we kindly ask the reviewer to refer to Subsection 2.4 and the part titled "The Role of G" on Page 6 of the revised manuscript. In addition to this, we have revised Subsection 2.4 to highlight the role of the feature weight matrix U and the representation matrix V, which play a major role in selecting the underlying features in our proposed method.
>
> For more information, we kindly ask the reviewer to refer to Subsection 2.4 on Page 6 of the revised manuscript, which involves the parts titled "The Role of U", "The Role of V", and "The Role of G."
>
> **(2) Discussion of Related Work:** We expanded our discussion of related work to provide a more comprehensive overview of existing methods in unsupervised feature selection. We included additional references and compared our method with these approaches in greater detail, highlighting the unique contributions and advantages of our proposed method.
>
> In response to this comment, we have added a new section titled "Related Work" to our paper. This section provides a detailed descriptions of the methods as outlined in Table 1. For more information, we kindly ask the reviewer to refer to Table 1 on Page 3 and Appendix A1 on Page 16 of the revised manuscript.
>
> **(3) Sensitivity Analysis:** We conducted a more thorough sensitivity analysis to address the reviewer's concerns about the method's sensitivity. This includes additional experiment and detailed discussions on how the choice of parameters and different dataset characteristics affect the performance of our method. The results and insights from this analysis have been added to the revised manuscript.
>
> In response to this comment, we have added a subsection titled "Sensitivity Analysis" in Appendix Subsection A7 on Page 20. Additionally, we have changed the title of the "Ablation Study" subsection in the main part of the manuscript to ``The Role of Hyperparameters." This subsection now includes both an ablation study and explanations about the sensitivity experiment results.
>
> We have carefully revised the manuscript according to the reviewer's suggestions and believe these changes have significantly improved the clarity and robustness of our paper. We are confident that the revised version addresses the concerns raised and meets the high standards of TMLR.
>
> Thank you once again for your guidance and consideration. We look forward to the final decision.